# Variational inverse modeling within the Community Inversion Framework v1.1 to assimilate $\delta^{13}\text{C(CH}_4)$ and $\text{CH}_4$ : a case study with model LMDz-SACS

Joël Thanwerdas[1,*], Marielle Saunois[1], Antoine Berchet[1], Isabelle Pison[1], Bruce H. Vaughn[2], Sylvia Englund Michel[2], and Philippe Bousquet[1]

[1]Laboratoire des Sciences du Climat et de l'Environnement, CEA-CNRS-UVSQ, IPSL, Gif-sur-Yvette, France.
[2]INSTAAR - University of Colorado, Boulder, CO, United States

**Correspondence:** J. Thanwerdas (joel.thanwerdas@lsce.ipsl.fr)

**Abstract.**

Atmospheric $\text{CH}_4$ mole fractions resumed their increase in 2007 after a plateau during the 1999-2006 period, indicating relative changes in the sources and sinks. Estimating sources by exploiting observations within an inverse modeling framework (top-down approaches) is a powerful approach. It is nevertheless challenging to efficiently differentiate co-located emission categories and sinks by using $\text{CH}_4$ observations alone. As a result, top-down approaches are limited when it comes to fully understanding $\text{CH}_4$ burden changes and attribute these changes to specific source variations. $\delta^{13}\text{C(CH4)}_{\text{source}}$ isotopic signatures of $\text{CH}_4$ sources differ between emission categories (biogenic, thermogenic and pyrogenic), and can therefore be used to address this limitation. Here, a new 3-D variational inverse modeling framework designed to assimilate $\delta^{13}\text{C(CH}_4)$ observations together with $\text{CH}_4$ observations is presented. This system is capable of optimizing both the emissions and associated source signatures of multiple emission categories at the pixel scale. To our knowledge, this represents the first attempt to carry out variational inversion assimilating $\delta^{13}\text{C(CH}_4)$ with a 3-D chemistry-transport model (CTM) and to independently optimize isotopic source signatures of multiple emission categories. We present the technical implementation of joint $\text{CH}_4$ and $\delta^{13}\text{C(CH}_4)$ constraints in a variational system, and analyze how sensitive the system is to the setup controlling the optimization using the LMDz-SACS 3-D CTM. We find that assimilating $\delta^{13}\text{C(CH}_4)$ observations and allowing the system to adjust isotopic source signatures provide relatively large differences in global flux estimates for wetlands ($-5.7 \text{ TgCH}_4.\text{yr}^{-1}$), agriculture and waste ($-6.4 \text{ TgCH}_4.\text{yr}^{-1}$), fossil fuels ($+8.6 \text{ TgCH}_4.\text{yr}^{-1}$) and biofuels-biomass burning ($+3.2 \text{ TgCH}_4.\text{yr}^{-1}$) categories compared to the results inferred without assimilating $\delta^{13}\text{C(CH}_4)$ observations. More importantly, when assimilating both $\text{CH}_4$ and $\delta^{13}\text{C(CH}_4)$ observations, but assuming that the source signatures are perfectly known, these differences increase by a factor of 3-4, strengthening the importance of having as accurate signature estimates as possible. Initial conditions, uncertainties in $\delta^{13}\text{C(CH}_4)$ observations or the number of optimized categories have a much smaller impact (less than $2 \text{ TgCH}_4.\text{yr}^{-1}$).

# 1 Introduction

Methane ($CH_4$) is a powerful greenhouse gas and is responsible for 23 % (Etminan et al., 2016) of the radiative forcing induced by the well-mixed greenhouse gases ($CO_2$, $CH_4$, $N_2O$). Atmospheric $CH_4$ mole fractions have increased quasi-continuously since the pre-industrial era and by about 9 $ppb.yr^{-1}$ from 1984 to 1998 (Dlugokencky, 2021). After a plateau between 1999 and 2006 that still generates attention and controversy (e.g., Fujita et al., 2020; Thompson et al., 2018; McNorton et al., 2018; Turner et al., 2017; Schaefer et al., 2016; Schwietzke et al., 2016; Rice et al., 2016), the mole fractions resumed their increase at a large rate, exceeding 10 $ppb.yr^{-1}$ in 2014 and 2015. Trends in atmospheric $CH_4$ are caused by a small imbalance between large sources and sinks. Assessing their spatio-temporal characteristics is particularly challenging considering the variety of $CH_4$ emissions. Yet, identifying and quantifying the processes contributing to these changes is mandatory to formulate relevant $CH_4$ mitigation policies that would contribute to meet the target of the 2015 UN Paris Agreement on Climate Change and to limit climate warming to $2\,°C$.

Thanks to continuous efforts of surface monitoring networks, the spatial coverage and the accuracy of the atmospheric $CH_4$ measurements provided to the scientific community increased over the last decades. Consequently, top-down estimates using inversion methods emerged and became relevant, along with bottom-up estimates, to explain and quantify the recent sources and sinks variations. The first inverse modeling techniques were designed in the late 1980s and early 1990s for inferring greenhouse gas sources and sinks from atmospheric $CO_2$ measurements (Enting and Newsam, 1990; Newsam and Enting, 1988). Without regularization of the problem, e.g. providing prior information, the inverse problem is ill-conditioned (or ill-posed). It means that there is no unique solution to the problem but also that a small error in the assimilated data (here atmospheric observations) can result in large errors in the derived solution. Several inversion methods have been designed over the years, among which analytical (e.g., Bousquet et al., 2006; Gurney et al., 2002), ensemble (e.g., Zupanski et al., 2007; Peters et al., 2005) and variational methods (e.g., Chevallier et al., 2005). The variational formulation uses the adjoint equations of a specific model to compute the gradient of a cost function and then minimize it, for example using a gradient descent method. Computational times and memory costs do not scale with the number of measurements and the number of variables to control, contrary to the analytical and ensemble methods, which can hardly accommodate very large observational datasets and control vectors at the same time. Thus, the variational formulation is preferred to the others when optimizing emissions and sinks at the pixel scale using large volumes of observational data, although its main limitation is the numerical cost to access posterior uncertainties when there is non-linearity in the inversion problem (Berchet et al., 2021).

Inversion systems generally assimilate measurements from ground-based stations and/or satellites to constrain the global sources and sinks of $CH_4$, starting from a prior knowledge of these. These systems are very effective to provide total emission estimates (e.g., Saunois et al., 2020; Bergamaschi et al., 2018, 2013; Saunois et al., 2017; Houweling et al., 2017, and references therein). However, differentiating the contributions of multiple co-located $CH_4$ source categories is challenging as it only relies on different seasonality cycles and on applied spatial distributions and error correlations (e.g., Bergamaschi et al., 2013, 2010). The atmospheric isotopic signal contains additional information on $CH_4$ emissions that can help to separate emission categories

based on their source origin. The atmospheric isotopic signal $\delta^{13}C(CH_4)$ is defined as:

$$\delta^{13}C(CH_4) = \frac{R}{R_{\text{std}}} - 1 \tag{1}$$

where R and $R_{\text{std}}$ denote the sample and standard $^{13}CH_4{:}^{12}CH_4$ ratios. We use the Vienna - Pee Dee Belemnite (V-PDB) scale with $R_{\text{std}}$ = 0.00112372 (Craig, 1957) throughout this paper. The isotopic source signatures of $CH_4$, here denoted by $\delta^{13}C(CH4)_{\text{source}}$, notably differ between emission categories ranging from $^{13}C$-depleted biogenic sources ($-61.7 \pm 6.2$ ‰, one standard deviation) and thermogenic sources ($-44.8 \pm 10.7$ ‰) to $^{13}C$-enriched thermogenic sources ($-26.2 \pm 4.8$ ‰) (Sherwood et al., 2017; Schwietzke et al., 2016), although the distributions are very large and overlaps exist between the extreme values. Consequently, $\delta^{13}C(CH_4)$ depends on both $CH_4$ emissions and their isotopic signatures. Saunois et al. (2017) pointed out that many emission scenarios inferred from atmospheric inversions are not consistent with $\delta^{13}C(CH_4)$ observations and that this constraint must be integrated into the inversion systems to avoid such inconsistencies. In addition, they highlighted the sensitivity of the atmospheric isotopic signal to the source partitioning and prescribed isotopic ratios. Since the 1990s, $\delta^{13}C(CH_4)$ has been monitored at multiple sites, providing opportunities to use this constraint within an inversion framework. In addition, these values have been shifting towards more negative values since 2006 (Nisbet et al., 2019) when $CH_4$ trends resumed their increase, suggesting that this isotopic data can help to understand the processes that contributed to the renewed growth. However, implementing the assimilation of such measurements into an inversion system is not straightforward and introduces additional complexity.

Hereinafter, the assimilation of $\delta^{13}C(CH_4)$ observations to constrain the estimates of an inversion is referred to as the "isotopic constraint". The implementation of such a constraint in an inversion system have already been attempted in previous studies focusing on $CH_4$ (e.g., Thompson et al., 2018; McNorton et al., 2018; Rigby et al., 2017; Rice et al., 2016; Schaefer et al., 2016; Schwietzke et al., 2016; Rigby et al., 2012; Neef et al., 2010; Bousquet et al., 2006; Fletcher et al., 2004) but, to our knowledge, never in a variational system associated to a 3-D chemistry-transport model (CTM). Adding this isotopic constraint to a variational inversion system is challenging as, in contrast to an analytic inversion in which the response functions of the model are precomputed, the isotopic constraints have to be considered both in the forward (simulated isotopic values) and the adjoint (sensitivity of isotopic observations to optimized variables) versions of the model.

This new system was implemented in the Community Inversion Framework (CIF), supported by the European Union H2020 project VERIFY (http://www.community-inversion.eu) and required to implement new forward, tangent-linear and adjoint operations. The forward operations were previously used to estimate the impact of the Cl sink on the modeling of $CH_4$ and $\delta^{13}C(CH_4)$ in LMDz-SACS (Thanwerdas et al., 2019). The purpose of this study is to present the technical implementation of the isotopic constraint in a variational inversion system and to investigate the sensitivity of this new configuration to different parameters. Our aim is not to estimate trends in sectoral emissions over the last two decades : future studies will address the estimation of $CH_4$ emissions over longer periods of time using this new system. The technical implementation and the various tested configurations are presented in Sect. 2. We analyze the results in Sect. 3. Sect. 4 presents our conclusions and recommendations on using such a multi-constraint variational system.

## 2 Methods

### 2.1 Theory of variational inversion

The notations introduced here follow the convention defined by Ide et al. (1997) and Rayner et al. (2019). The observation vector is called $\mathbf{y}^o$. It includes here all available observations, namely $CH_4$ and $\delta^{13}C(CH_4)$ measurements retrieved by surface stations, over the full simulation time-window (see Sect. 2.4.2). The associated errors are assumed to be unbiased and Gaussian and are described within the error covariance matrix $\mathbf{R}$. This matrix accounts for all errors contributing to mismatches between simulated and observed values. $\mathbf{x}$ is the control vector and includes all the variables (here $CH_4$ surface fluxes, initial $CH_4$ mole fractions, source signatures $\delta^{13}C(CH4)_{source}$ and initial $\delta^{13}C(CH_4)$ values) optimized by the inversion system. Hereinafter, these variables will be referred to as the "control variables". Prior information about the control variables are provided by the vector $\mathbf{x}^b$. Its associated errors are also assumed to be unbiased and Gaussian and are described within the error covariance matrix $\mathbf{B}$. $\mathcal{H}$ is the observation operator that projects the control vector $\mathbf{x}$ into the observation space. This operator mainly consists of the 3-D CTM (here LMDz-SACS introduced in Sect 2.2). Nevertheless, the CTM is followed by spatial and time operators, which interpolate the simulated fields to produce simulated equivalents of the assimilated observations at specific locations and times, making the simulations and observations comparable. An additional 'transformation' operator, implemented in the new system, enables comparison between distinct simulated tracers, e.g., $^{12}CH_4$ and $^{13}CH_4$, and observations, e.g., $\delta^{13}C(CH_4)$ (see Sect 2.3).

In a variational formulation of the inference problem that allows for $\mathcal{H}$ non-linearity, the cost function $J$ is defined as :

$$J(\mathbf{x}) = \frac{1}{2}(\mathbf{x} - \mathbf{x}^b)^{\mathrm{T}}\mathbf{B}^{-1}(\mathbf{x} - \mathbf{x}^b) + \frac{1}{2}(\mathcal{H}(\mathbf{x}) - \mathbf{y}^o)^{\mathrm{T}}\mathbf{R}^{-1}(\mathcal{H}(\mathbf{x}) - \mathbf{y}^o) \tag{2}$$

$$= J_b(\mathbf{x}) + J_o(\mathbf{x}) \tag{3}$$

The cost function is therefore a sum of two parts :

- The first part is induced by the differences between the posterior and prior variables ($J_b$).

- The second is induced by the differences between simulations and observations ($J_o$)

The minimum of $J$ can be reached iteratively with a descent algorithm that requires several computations of the gradient of $J$ with respect to the control vector $\mathbf{x}$:

$$\nabla J_{\mathbf{x}} = \mathbf{B}^{-1}(\mathbf{x} - \mathbf{x}^b) + \mathcal{H}^*(\mathbf{R}^{-1}(\mathcal{H}(\mathbf{x}) - \mathbf{y}^o)) \tag{4}$$

$\mathcal{H}^*$ denotes the adjoint operator of $\mathcal{H}$. As in analytical and ensemble methods, the variational formulation necessitates the inversion of both error matrices $\mathbf{R}$ and $\mathbf{B}$. In most applications, $\mathbf{R}$ is considered diagonal as point observations are distant in time and space (i.e., uncorrelated observation errors), allowing for the inverse to be calculated easily, although that assumption should be revised with the increasing availability of satellite sources (Liu et al., 2020). B is rarely diagonal due to spatial and temporal correlations of errors in the fluxes. However, B is often decomposed as combinations of smaller matrices, e.g., using Kronecker products of sub-correlation matrices, which allows to compute its inverse by blocks.

## 2.2 The Chemistry-Transport Model

The LMDz general circulation model (GCM) is the atmospheric component of the Institut Pierre-Simon Laplace Coupled Model (IPSL-CM) developed at the Laboratoire de Météorologie Dynamique (LMD) (Hourdin et al., 2006). The version of LMDz we use is an 'offline' version dedicated to the inversion framework created by Chevallier et al. (2005): precomputed air mass fluxes provided by the online version of LMDz are given as inputs to the transport model, reducing significantly the computational time. The model is set up at a horizontal resolution of $3.8° \times 1.9°$ (96 grid cells in longitude and latitude) with 39 hybrid sigma-pressure levels reaching an altitude up to about 75 km. About 20 levels are dedicated to the stratosphere and the mesosphere. The model time-step is 30 min and the output mole fractions are 3-hourly snapshots. The horizontal winds are nudged towards ECMWF meteorological analyses (ERA-Interim) in the online version of the model then fed to the offline version. Vertical diffusion is parameterized by a local approach from Louis (1979), and deep convection processes are parameterized by the Tiedtke (1989) scheme.

The offline model LMDz is coupled with the Simplified Atmospheric Chemistry System (SACS) (Pison et al., 2009). This chemistry system was previously used to simulate the oxidation chain of hydrocarbons, including $CH_4$, formaldehyde ($CH_2O$), carbon monoxide (CO) and molecular hydrogen ($H_2$). For the purpose of this study, this system was converted into a chemistry parsing system. It follows the same principle as the one used by the regional model CHIMERE (Menut et al., 2013) and therefore allows for user-specific chemistry reactions. As a result, it generalizes the previous SACS module to any possible set of reactions. The adjoint code has also been implemented to allow variational inverse modeling. The different species are either prescribed (here OH, O($^1$D) and Cl) or simulated (here $^{12}CH_4$ and $^{13}CH_4$). The prescribed species are not transported in LMDz, nor are their mole fractions updated through chemical production or destruction. Such species are only used to calculate reaction rates to update simulated species at each model time step. In this study, the isotopologues $^{12}CH_4$ and $^{13}CH_4$ are simulated as separate tracers and $CH_4$ is defined as the sum of both isotopologues. Cl + $CH_4$ oxidation was implemented to complete the chemical removal of $CH_4$, which previously only accounted for OH + $CH_4$ and O($^1$D) + $CH_4$ in the SACS scheme.

In the atmosphere, radicals (OH, O($^1$D) or Cl) react faster with $^{12}CH_4$ than with $^{13}CH_4$. This effect is called the Kinetic Isotope Effect (KIE) or the fractionation effect. Fractionation values are prescribed to the different sinks in SACS. Here, this value is defined by $KIE = k_{12}/k_{13}$ where $k_{12}$ is the rate constant of a reaction between a radical and $^{12}CH_4$. $k_{13}$ is the rate constant of the reaction between the same radical and $^{13}CH_4$. Additional information and prescribed KIE values are provided in the supplement (Text S2).

The chemistry-transport LMDz-SACS is used to test the new variational inverse modeling system that is described in the next section.

## 2.3 Technical implementation of the isotopic constraint

The isotopic multi-constraint system was implemented in the CIF. The CIF has been designed to allow comparison of different approaches, models and inversion systems used in the inversion community (Berchet et al., 2021). Different atmospheric

transport models, regional and global, Eulerian and Lagrangian are implemented within the CIF. The system presented in this paper has been originally designed to run and be tested with LMDz-SACS but can theoretically be coupled with all models implemented in the CIF framework. The system is able to :

- Assimilate $\delta^{13}C(CH_4)$ and $CH_4$ observations together.

- Independently optimize fluxes and isotopic signatures for multiple emission categories.

- Optimize $\delta^{13}C(CH_4)$ and $CH_4$ initial conditions.

Figure 1 shows the different steps of a minimization iteration of the cost function. Each iteration performed with the descent algorithm can be decomposed into four main steps presented below. For clarity, we only present here the optimization of $CH_4$ fluxes and associated source signatures but $CH_4$ and $\delta^{13}C(CH_4)$ initial conditions can also be optimized by the system following the same process.

1. The process starts with a forward run. The different flux variables are extracted and converted into $^{12}CH_4$ and $^{13}CH_4$ mass fluxes for each category following the Eq. 5 and 6 below.

$$F_{12}^i = \frac{M_{12}}{M_T} \cdot \frac{1}{1 + A^i} \cdot F_T^i \tag{5}$$

$$F_{13}^i = \frac{M_{13}}{M_T} \cdot \frac{A^i}{1 + A^i} \cdot F_T^i \tag{6}$$

with

$$A^i = (1 + \delta^{13}C(CH4)_{\text{source}}^i) \cdot R_{\text{std}} \tag{7}$$

$F_T^i$, $F_{12}^i$ and $F_{13}^i$ are the $CH_4$, $^{12}CH_4$ and $^{13}CH_4$ mass fluxes of a specific category $i$, respectively. $M_T$, $M_{12}$ and $M_{13}$ are the $CH_4$, $^{12}CH_4$ and $^{13}CH_4$ molar masses, respectively. $\delta^{13}C(CH4)_{\text{source}}^i$ is the isotopic signature of the category $i$. $M_T$ should preferably depend on $M_{12}$ and $M_{13}$ when converting the mass fluxes:

$$M_T = \frac{M_{12} + A^i \cdot M_{13}}{1 + A^i} \tag{8}$$

However, the complexity of the forward, tangent-linear and adjoint codes would be largely enhanced by such a relationship. The code structure would also be less generic, i.e., it could not be used for a joint assimilation of multiple isotopologues of $CH_4$, such as both $\delta^{13}C(CH_4)$ and $\delta D(CH_4)$. We choose to implement $M_T$ as a constant that can be prescribed freely by the user, therefore without considering any influence of the $M_{12}$ and $M_{13}$ values, also prescribed by the user. As the observed isotopic source signatures roughly vary between $-70\,‰$ and $-10\,‰$, a maximum variation of 0.004 % in $M_T$ could be expected. It will very likely not affect the results of our study or that of any other inversion performed with our system.

The $^{12}CH_4$ and $^{13}CH_4$ total fluxes are then calculated by summing all categories and used by the model LMDz-SACS to simulate the $^{12}CH_4$ and $^{13}CH_4$ atmospheric mole fractions over the time-window considered. After the simulation,

the simulated values are converted to $CH_4$ and $\delta^{13}C(CH_4)$ simulated equivalent of the assimilated observations using Eq. 9 and 10 below :

$$[CH_4] = [^{12}CH_4] + [^{13}CH_4] \tag{9}$$

$$\delta^{13}C(CH_4) = \frac{[^{13}CH_4]}{[^{12}CH_4]} \cdot \frac{1}{R_{std}} - 1 \tag{10}$$

$[CH_4]$, $[^{12}CH_4]$ and $[^{13}CH_4]$ are $CH_4$, $^{12}CH_4$ and $^{13}CH_4$ atmospheric mole fractions simulated by the model in $\mathrm{mol\,mol^{-1}}$, respectively.

2. These simulated values are then compared to the available observations in order to compute $\mathcal{H}(\mathbf{x}) - \mathbf{y}^o$ which is further used to infer the cost function and generate $CH_4$ and $\delta^{13}C(CH_4)$ adjoint forcings (indicated by the "*" star superscript symbol) that compose the vector $\delta\mathbf{y}^*$:

$$\delta\mathbf{y}^* = \mathbf{R}^{-1}(\mathcal{H}(\mathbf{x}) - \mathbf{y}^o) \tag{11}$$

Although this vector is normally used directly as input to the adjoint model (see Eq. 4), the $CH_4$ and $\delta^{13}C(CH_4)$ adjoint forcings must first be converted into the $^{12}CH_4$ and $^{13}CH_4$ adjoint forcings in the new system.

3. The newly designed adjoint code that converts $CH_4$ and $\delta^{13}C(CH_4)$ adjoint forcings into $^{12}CH_4$ and $^{13}CH_4$ adjoint forcings is based on the Eq. 12, 13 and 14 depending on the type of the initial observation.

$$[^{12}CH_4]^*_{CH_4} = [^{13}CH_4]^*_{CH_4} = [CH_4]^* \tag{12}$$

$$[^{12}CH_4]^*_{\delta^{13}C} = -\frac{[^{13}CH_4]}{[^{12}CH_4]^2} \cdot \frac{1}{R_{std}} \cdot \delta^{13}C(CH_4)^* \tag{13}$$

$$[^{13}CH_4]^*_{\delta^{13}C} = \frac{1}{[^{12}CH_4]} \cdot \frac{1}{R_{std}} \cdot \delta^{13}C(CH_4)^* \tag{14}$$

$[^{12}CH_4]^*_{CH_4}$ and $[^{13}CH_4]^*_{CH_4}$ are adjoint forcings associated with $CH_4$ observations. $[^{12}CH_4]^*_{\delta^{13}C}$ and $[^{13}CH_4]^*_{\delta^{13}C}$ are adjoint forcings associated with $\delta^{13}C(CH_4)$ observations. The adjoint code of the CTM is then run with these adjoint forcings as inputs.

Outputs of the adjoint run provide the sensitivities of the adjoint forcings to the $^{12}CH_4$ and $^{13}CH_4$ mass fluxes of a specific category $i$, denoted by $F_{12}^{*,i}$ and $F_{13}^{*,i}$ . Equations 15 and 16 convert them back to sensitivities to the initial control variables, denoted by $F_T^{*,i}$ and $\delta^{13}C(CH4)_{source}^{*,i}$.

$$F_T^{*,i} = \frac{1}{1+A} \cdot [\frac{M_{12}}{M_T} \cdot F_{12}^{*,i} + \frac{M_{13}}{M_T} \cdot A \cdot F_{13}^{*,i}] \tag{15}$$

$$\delta^{13}C(CH4)_{source}^{*,i} = R_{std} \cdot \frac{F_T}{(1+A)^2} \cdot [\frac{M_{13}}{M_T} \cdot F_{13}^{*,i} - \frac{M_{12}}{M_T} \cdot F_{12}^{*,i}] \tag{16}$$

4. The minimization algorithm utilizes these sensitivities to compute the gradient of the cost function. It then finds an optimized control vector reducing the cost function and used for the next iteration.

In order to confirm that the several adjoint operations have been correctly implemented, we also provide the results of multiple adjoint tests in the supplement (Text S4).

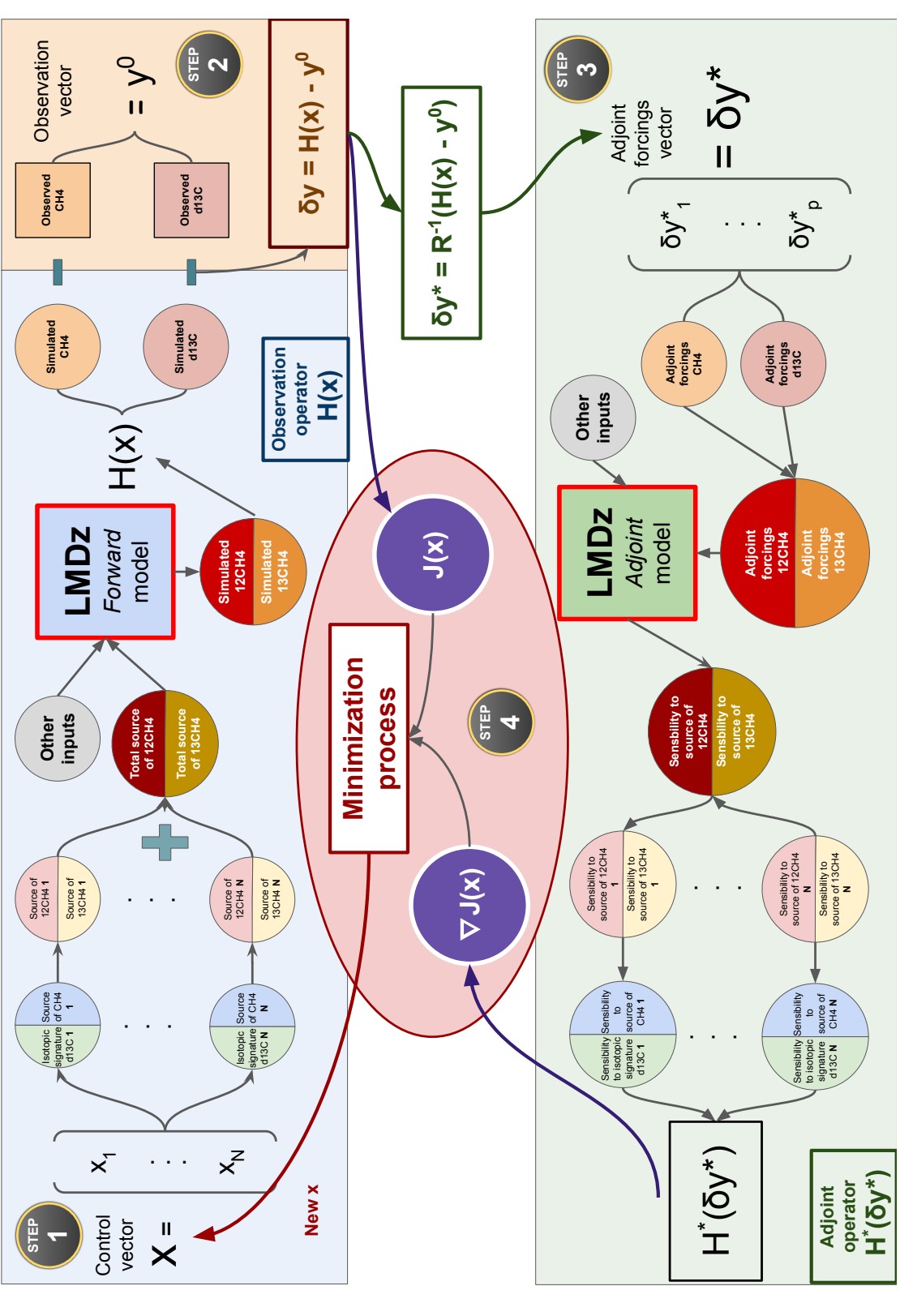

**Figure 1.** The minimization iteration process in the newly designed system.

]The minimization iteration process in the newly designed system. The "step" black circles with a gold border indicates the reading direction to follow. Step 1 (blue rectangle) refers to a forward run. Step 2 (orange rectangle) refers to the forward and adjoint operations required to compare observations and simulated values. Step 3 (green rectangle) refers to an adjoint run. This step must be read from the right to the left. Step 4 (red ellipse) refers to the minimization of the cost function operated by the dedicated minimization algorithm. Note that results of Step 2 are used both in the minimization process (red ellipse) and as inputs for Step 3. The minimization iteration process is also illustrated in the supplement (Fig. S1)

## 2.4   Setup of the reference simulation

The reference configuration (REF) is a variational inversion that optimizes the $CH_4$ emission fluxes and $\delta^{13}C(CH_4)$ isotopic source signatures of five different categories : biofuels-biomass burning (BB), agriculture and waste (AGW), fossil fuels (FF), wetlands (WET) and other natural sources (NAT). $CH_4$ and $\delta^{13}C(CH_4)$ initial conditions are also optimized. The assimilation time-window is the 2012-2017 period. The five categories originate from an aggregation of ten sub-categories (see Table 1) and are chosen to be as isotopically consistent as possible. Sinks are not optimized here.

### 2.4.1   Control vector x and B matrix

**Table 1.** Emissions and flux-weighted isotopic signatures of the $CH_4$ sources averaged over 2012-2017 for different categories and their sub-categories. Prior uncertainties in fluxes are set to 100 % for all categories and sub-categories. * Unc. : Prior uncertainty in the isotopic signature prescribed to the category or the sub-category as a percentage of the signature. CH19 : Chang et al. (2019) ; GA18 : Ganesan et al. (2018) ; SH17 : Sherwood et al. (2017) ; WA16 : Warwick et al. (2016) ; ZA16 : Zazzeri et al. (2016) ; TO12 : Townsend-Small et al. (2012) ; KL10 : Klevenhusen et al. (2010) ; BO06 : Bousquet et al. (2006) ; BR01 : Bréas et al. (2001) ; SA01 : Sansone et al. (2001) ; CH00 : Chanton et al. (2000) ; HO00 : Holmes et al. (2000) ; CH99 : Chanton et al. (1999) ; BE98 : Bergamaschi et al. (1998) ; LE93 : Levin et al. (1993).

| Categories | Emissions (Tg.yr$^{-1}$) | Signature (‰) | Unc.* (%) | Sub-categories | Emissions (Tg.yr$^{-1}$) | Signature (‰) | Unc.* (%) | Signature references |
|---|---|---|---|---|---|---|---|---|
| WET | 180.3 | -60.8 | 20 | Wetlands | 180.3 | -60.8 | 20 | GA18 |
| AGW | 226.4 | -59.1 | 20 | Rice cultivation | 38.0 | -63 | 20 | SH17 ; BO06 ; BR01 |
| | | | | Livestock | 117.8 | -63.6 | 20 | CH19 |
| | | | | Waste | 70.6 | -49.5 | 20 | KL10 ; TO12 ; CH99 ; BE98 ; LE93 |
| FF | 116.3 | -43.4 | 25 | Coal | 38.4 | -40.4 | 25 | SH17 ; ZA16 |
| | | | | Oil, Gas, Industry | 77.9 | -44.9 | 25 | SH17 |
| BB | 28.4 | -22.5 | 40 | Biofuels-biomass burning | 28.4 | -22.5 | 40 | BO06 ; CH00 |
| NAT | 38.1 | -49.9 | 15 | Oceanic sources | 14.4 | -42.0 | 20 | BR01 ; HO00 ; SA01 |
| | | | | Termites | 8.7 | -63.0 | 20 | TH18 ; SH17 ; WA16 |
| | | | | Geological (onshore) | 15.0 | -50.0 | 20 | BO06 |
| Total | 589.5 | -54.1 | | Total | 589.5 | -54.1 | | |

We adopt the $CH_4$ emissions compiled for inversions performed as part of the Global Methane Budget (Saunois et al., 2020). Anthropogenic (including biofuels) and biomass burning emissions are based on the EDGARv4.3.2 database (Janssens-Maenhout et al., 2017) and the GFED4s databases (van der Werf et al., 2017), respectively. Statistics from British Petroleum (BP) and the Food and Agriculture Organization of the United Nations (FAO) have been used to extend the EDGARv4.3.2

database, ending 2012, until 2017. The natural sources emissions are based on averaged literature values : Poulter et al. (2017) for wetlands, Kirschke et al. (2013) for termites, Lambert and Schmidt (1993) for oceanic sources and Etiope (2015) for geological (onshore) sources. Emissions from geological sources have been scaled down to 15 $TgCH_4.yr^{-1}$ in the prior

emissions adopted in Saunois et al. (2020). All prior fluxes are prescribed at monthly resolution and at the spatial resolution of LMDz. Globally-averaged emissions over the 2012-2017 period are listed in Table 1.

Prior estimates of isotopic source signatures are provided either at the pixel scale (for wetlands), at the regional scale based on TransCom regions (Patra et al., 2011) or at the global scale. The wetlands signature map is taken from Ganesan et al. (2018). Livestock isotopic source signatures are taken from Chang et al. (2019) and aggregated into the 11-regions map by selecting region-specific values. Livestock source signatures have been likely decreasing over time since the 1990s due to changes in C3/C4 diet within the major livestock producing countries and therefore annual values are prescribed. However, these estimates end in 2013 and we set the years 2014 to 2017 equal to the year 2013. Consequently, only the year 2012 has a different prescribed value from the other years. Coal and Oil, Gas, Industry (OGI) isotopic signature values are inferred from Sherwood et al. (2017) and Zazzeri et al. (2016) and aggregated into the same 11-regions map. The EDGARv4.3.2 categories PRO_OIL and PRO_GAS (fugitive emissions during oil and gas exploitation) largely contribute ($\sim$90 %) to the total of the "Oil, Gas & Industry" sub-category. Therefore, we chose to neglect the influence of other subsub-categories (such as industry) on the isotopic signature of the category. As for the biofuels-biomass burning category, we use region-specific signatures over 11 regions. A global signature value is prescribed for each of the other categories. Except for the livestock category, all prior signatures are set constant over time. To infer the $\delta^{13}C(CH4)_{source}$ map of a category based on the sub-categories, the $^{12}CH_4$ and $^{13}CH_4$ fluxes for each emission sub-category within a category are derived based on Eq. 5 and 6 and added up. The resulting fluxes are then converted back to a $\delta^{13}C(CH4)_{source}$ map representing the aggregated isotopic signature of the category. Additional information regarding the chosen isotopic signatures and their references is provided in the supplement (Text S1).

Three values per month (10 days, 10 days and the rest) for fluxes and their associated isotopic signatures are included in the control variables. Although the time variations of isotopic signatures are poorly constrained in the literature, we choose to include the same number of variables for fluxes and isotopic signatures in order to illustrate the full capabilities of the system and have it ready when more isotopic constraints will appear.

The portion of the diagonal of $\mathbf{B}$ associated to prior $CH_4$ emission fluxes is filled in with the variances set to 100 % of the square of the maximum of emissions over the cell and its eight neighbours during each month. Off diagonal terms of $\mathbf{B}$ (covariances) are based on correlation e-folding lengths (500 km over land and 1000 km over sea). The same method is applied for isotopic source signatures, although a specific percentage of uncertainties deduced from the global values of Sherwood et al. (2017) is used to infer each category diagonal term (see Table 1). No temporal correlations are considered here. Finally, prior uncertainties in initial conditions are set to 10 % for $CH_4$ ($\sim$ 180 ppb) and 3 % for $\delta^{13}C(CH_4)$ ($\sim$ 1.4 ‰).

### 2.4.2 Observation vector y and R matrix

$CH_4$ observations are taken from the data archived at the World Data Centre for Greenhouse Gases (WDCGG) of the WMO Global Atmospheric Watch (WMO-GAW) program. We selected 66 stations from 13 surface monitoring networks providing in-situ measurements of $CH_4$ mole fractions. The stations are displayed in Fig. 2.

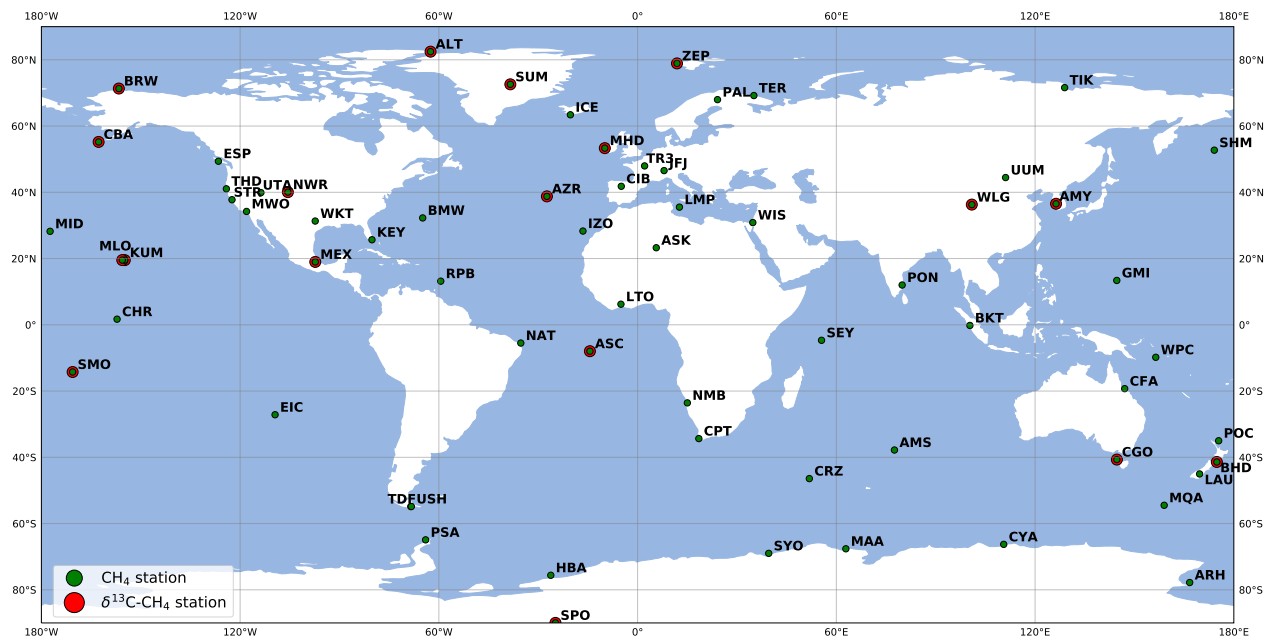

**Figure 2.** Locations of $CH_4$ and $\delta^{13}C(CH_4)$ surface stations. Affiliated networks are not displayed. More information can be found in the supplement (Table S3 and S4).

$\delta^{13}C(CH_4)$ observations are taken from 18 surface stations from the Global Greenhouse Gas Reference Network (GGGRN), part of the NOAA-ESRL's Global Monitoring Laboratory (NOAA GML). Air samples were collected on an approximately weekly basis during the 2012-2017 period and analyzed by the Institute of Arctic and Alpine Research (INSTAAR) to provide $\delta^{13}C(CH_4)$ isotope ratio measurements. The analytical uncertainty of the isotopic measurements, based on a surveillance cylinder, is 0.06 ‰. In this study, we focused on estimating monthly and annual flux variations rather than investigating daily or weekly variations. Prescribing error correlations in the **R** matrix (introduced in Sect. 2.1) can be used to ensure that the inversion preferentially constrains the components we are interested in (i.e., long-term trend and seasonal cycle). In order to keep the R matrix diagonal and to focus on monthly and annual variations of the signal, we chose to use $\delta^{13}C(CH_4)$ observational data based on a curve fitting the original $\delta^{13}C(CH_4)$ observations. The fitting curve is a function including 3 polynomial parameters (quadratic) and 8 harmonic parameters as in Masarie and Tans (1995). After the fitting, the pseudo-observations were sampled at the same time as the original observations. We also hypothesized that the convergence would be slightly faster if a smooth curve fitting the real observations was used instead of the real observations, which appeared to be false (see Sect. 3.1). One sensitivity inversion aims at estimating the error introduced by this simplification (simulation S2 in Table 2).

The **R** matrix for both $CH_4$ and $\delta^{13}C(CH_4)$ is defined as diagonal, assuming that observation errors are not correlated, neither in space nor in time. This diagonal matrix can be decomposed into two parts : measurement and model error variances.

Measurement errors account for instrumental errors whereas model errors encompass transport and representativity errors induced by the model :

$$\mathbf{R} = \mathbf{R_{measurement}} + \mathbf{R_{model}} \tag{17}$$

Here, we use the provided observation errors to fill the $\mathbf{R_{measurement}}$ diagonal matrix. Globalview-$CH_4$ (GLOBALVIEW-CH4, 2009) values are used to represent model errors and prescribe variances at each station for $CH_4$ mixing ratio measurements in order to fill the $\mathbf{R_{model}}$ diagonal matrix. This simple approach has been used previously in atmospheric inversions (Locatelli et al., 2015, 2013; Yver et al., 2011; Bousquet et al., 2006; Rodenbeck et al., 2003). Errors in Globalview-$CH_4$ are computed at each site as the Root-Mean-Square-Error (RMSE) of the measurements on a smooth curve fitting them. As

Globalview-$CH_4$ does not provide errors for Globalview-$CH_4$ measurements, the same method has been applied here. RMSE of the measurements on a smooth curve fitting them over the 2012-2017 period is prescribed as the standard deviation for each site providing $\delta^{13}C(CH_4)$ measurements. These errors range between 3-19 ppb for $CH_4$ observations and 0.11-0.20 ‰ for $\delta^{13}C(CH_4)$ observations. Mean prescribed errors for each station are provided in the supplement (Tables S3 and S4).

### 2.4.3   Spin-up

Before starting the inversion, the model has been spun-up during 30 years using constant emissions and recycling meteorology from the year 2012 in order to consider the long timescales for isotopic changes (Tans, 1997). At the end of the spin-up, $\delta^{13}C(CH_4)$ values have been offset (+1.4 ‰) to fit the global mean $\delta^{13}C(CH_4)$ in January 2012 and $CH_4$ mole fractions have been scaled to fit the global mean $CH_4$ mole fraction in January 2012. Due to the non-linearity of transport and mixing, offsetting $\delta^{13}C(CH_4)$ initial values in a forward run can generate errors. This impact is discussed later using a configuration

where $\delta^{13}C(CH_4)$ initial conditions have not been offset (S1).

### 2.4.4   Sensitivity tests

A set of nine different configurations, including REF, has been designed to assess the impact of assimilating $\delta^{13}C(CH_4)$ observations in addition to $CH_4$ observations and also to evaluate the sensitivity of the inversion results to the system's setup.
   Multiple parameters have been tested throughout the various configurations :

1. NOISO has no isotopic constraint. Therefore, this configuration only simulates $CH_4$ and assimilates $CH_4$ observations.

   2. S1 uses $\delta^{13}C(CH_4)$ initial conditions that are not offset and are therefore directly taken from the spin-up.

   3. S2 assimilates the real $\delta^{13}C(CH_4)$ observations instead of the fitting curve data.

   4. In S3, the $\delta^{13}C(CH_4)$ model uncertainties are divided by a factor 2.

   5. T1 uses 10 sub-categories instead of 5 aggregated categories, increasing the degrees of freedom.

**Table 2.** Nomenclature and characteristics of the configurations. Details are provided in Sect. 2.4.4. ** Prior uncertainties in initial $\delta^{13}C(CH_4)$ conditions have been set to 10 %.

| Name | $\delta^{13}C(CH_4)$ initial cond. | $\delta^{13}C(CH_4)$ observations | $\delta^{13}C(CH_4)$ model errors | $\delta^{13}C(CH4)_{source}$ regional variability | $\delta^{13}C(CH4)_{source}$ uncertainties | Number of categories |
|---|---|---|---|---|---|---|
| NOISO | Without isotopic constraint | | | | | 5 |
| REF | Offset | Curve fitting | RMSE obs-fit | Regional variability | REF uncertainties | 5 |
| S1 | No offset** | Curve fitting | RMSE obs-fit | Regional variability | REF uncertainties | 5 |
| S2 | Offset | Real obs. | RMSE obs-fit | Regional variability | REF uncertainties | 5 |
| S3 | Offset | Curve fitting | RMSE obs-fit / 2 | Regional variability | REF uncertainties | 5 |
| T1 | Offset | Curve fitting | RMSE obs-fit | Regional variability | REF uncertainties | 10 |
| T2 | Offset | Curve fitting | RMSE obs-fit | Global mean | REF uncertainties | 5 |
| T3 | Offset | Curve fitting | RMSE obs-fit | Regional variability | 1 % for each cat. | 5 |
| T4 | Offset | Curve fitting | RMSE obs-fit | Global mean | 1 % for each cat. | 5 |

6. In theory, the system is capable of optimally adjusting two source signatures if the assimilated information is sufficient. For instance, the system can choose to shift one signature downward and another upward in a given pixel, in order to improve the fitting in this specific pixel. The configuration T2 has been specifically designed to investigate whether the system would be able to retrieve a realistic distribution (similar to REF) starting from globally-averaged signatures for each category.

7. In T3, the $\delta^{13}C(CH_4)$ source signatures uncertainties are set to a very low value (1 %) in order to prevent the system from optimizing them. In other words, all changes are put on $CH_4$ emissions.

8. Finally, T4 applies both changes from T2 and T3.

Table 2 summarizes the different configurations and the associated changes. The configurations have been grouped into two sets to facilitate the analysis of results : on the one hand, S-group configurations (REF + S1-S4) have setup variations that are not expected to largely influence the results compared to REF. On the other hand, T-group configurations (T1-T4) alter parameters that are very likely to impact the results.

# 3 Results

## 3.1 Minimization of the cost function

The minimization process is performed using the M1QN3 algorithm (Gilbert and Lemaréchal, 1989). One full simulation (forward + adjoint) with the isotopic constraint necessitates about 170 CPU hours to run 6 years, i.e., 2.4 CPU hours per month simulated. The computational burden is increased by a factor 2 in comparison to an inversion without the isotopic

constraint due to the doubling of simulated tracers ($^{12}CH_4$ and $^{13}CH_4$). One full simulation is generally enough to complete one iteration of the minimization process but two or three simulations are sometimes required by M1QN3. Therefore, the number of simulations is slightly larger than the number of iterations. Figure 3 displays the minimization process of the cost function for all configurations.

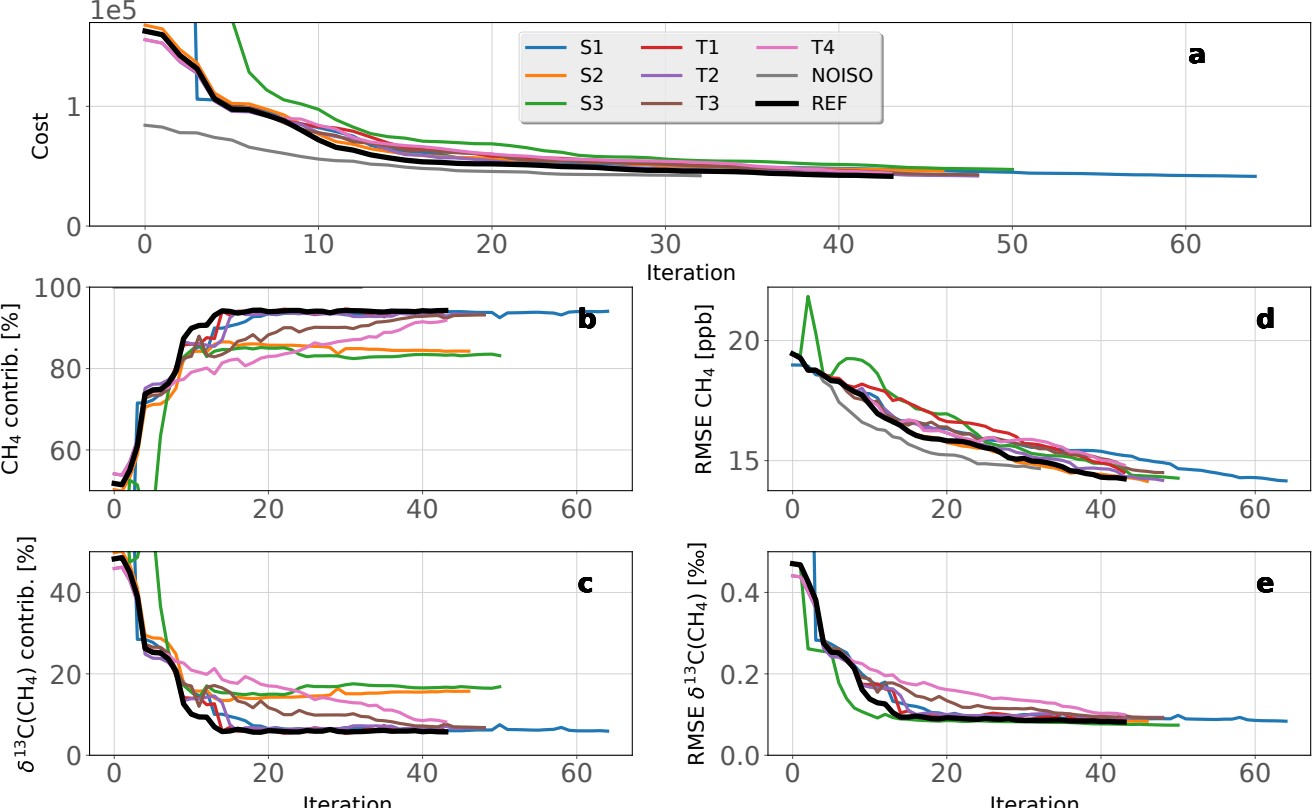

**Figure 3.** Minimization of the cost function for all configurations. a) Value of the cost function with respect to the number of iterations. b) $CH_4$ contribution to $J_o$. c) $\delta^{13}C(CH_4)$ contribution to $J_o$. d) RMSE associated to observed-simulated $CH_4$. e) RMSE associated to observed-simulated $\delta^{13}C(CH_4)$. For clarity reasons, S1 and S3 initial values are not displayed because they are much larger than those of REF.

Except for S1 and T1, the inversions were stopped when the gradient norm reduction exceeded 96 % for the third consecutive iteration. Number of iterations are compared to investigate the sensitivity of the computational cost to the setup. 32 iterations (37 simulations) for NOISO, 43 iterations (47 simulations) for REF and about 50 iterations for the others were necessary. Consequently, although assimilating $\delta^{13}C(CH_4)$ observations requires at least 11 additional iterations, the setup has little

influence on the number of iterations if the same convergence criteria is used. Also, using curve-fitted data instead of real observations do not reduce the computational burden as we first speculated.

S1 and T1 inversions were extended until their cost function reached the same reduction as REF in order to estimate the additional computational burden required to reach similar results when initial conditions are not offset (S1) and the number of categories is increased (T1). 10 and 21 additional iterations were necessary for T1 and S1, respectively. For T1, it shows

that increasing the degrees of freedom also increases the computational burden. For S1, it highlights the benefits of offsetting $\delta^{13}C(CH_4)$ initial conditions.

As we assume no correlation of errors in R, $J_o$ (see Eq. 3) can be divided into $CH_4$ and $\delta^{13}C(CH_4)$ contributions. Figure 3 shows that all configurations lead to a fast reduction of the $\delta^{13}C(CH_4)$ contribution. During the first ten iterations, it decreased from 50-90 % (depending on the configuration) to 10-20 %. Conversely, the $CH_4$ contribution increased from 10-50 % to

80-90 %. By adjusting the isotopic source signatures (all configurations besides T3-T4), the system was able to efficiently and rapidly reduce the discrepancies between simulated and observed $\delta^{13}C(CH_4)$. As a result, the $\delta^{13}C(CH_4)$ RMSE decreased very rapidly during the first ten iterations while the $CH_4$ RMSE decreased at a roughly constant rate. Consequently, the system is preferentially adjusting $\delta^{13}C(CH_4)$ over $CH_4$ values to reduce the cost function, presumably because the ratio of RMSE to prescribed observational error for $\delta^{13}C(CH_4)$ is, on average, about twice as large as for $CH_4$. In other terms, it is simpler

for the system to adjust $\delta^{13}C(CH_4)$ before attempting to modify $CH_4$. The ratio of the number of $\delta^{13}C(CH_4)$ observations to the number of $CH_4$ observations is not expected to play a significant role in the convergence process, although we did not rigorously study this influence. This ratio is only expected to affect the contribution of a component ($\delta^{13}C(CH_4)$ or $CH_4$) to the total cost function.

The decrease rate associated with $\delta^{13}C(CH_4)$ RMSE can be increased by reducing the model uncertainties prescribed to

the $\delta^{13}C(CH_4)$ observations. S3 is an example of such an adjustment, as the model uncertainties have been divided by two. With this configuration, the system requires five less iterations than REF to reach a similar $\delta^{13}C(CH_4)$ RMSE reduction but 7 additional iterations to reach a similar $CH_4$ RMSE reduction. T3 and T4 configurations constrain the isotopic signatures, thus the reduction of the $\delta^{13}C(CH_4)$ contribution necessitates 25 more iterations than REF to reach similar RMSE reduction. To summarize, the decrease rate associated with $\delta^{13}C(CH_4)$ RMSE is highly dependent on the prescribed uncertainties in

$\delta^{13}C(CH_4)$ observations and the ability of the system to adjust source signatures.

## 3.2   $CH_4$ and $\delta^{13}C(CH_4)$ fitting

As expected, the assimilation process greatly improves the agreement between simulated and observed values for both $CH_4$ and $\delta^{13}C(CH_4)$. Figure 4 shows the globally-averaged time-series of $CH_4$ and $\delta^{13}C(CH_4)$.

$CH_4$ RMSE using prior estimates is 19.4 ppb and drops to 14.3 $\pm$ 0.2 ppb ($1\sigma$) on average over all the configurations

using posterior estimates. Prior estimates capture well the observed $CH_4$ and the improvement is therefore relatively small. In addition, all configuration results regarding $CH_4$ are very similar. In particular, NOISO is not performing much differently than the other configurations, indicating that the additional isotopic constraint does not affect the fitting to $CH_4$ observations.

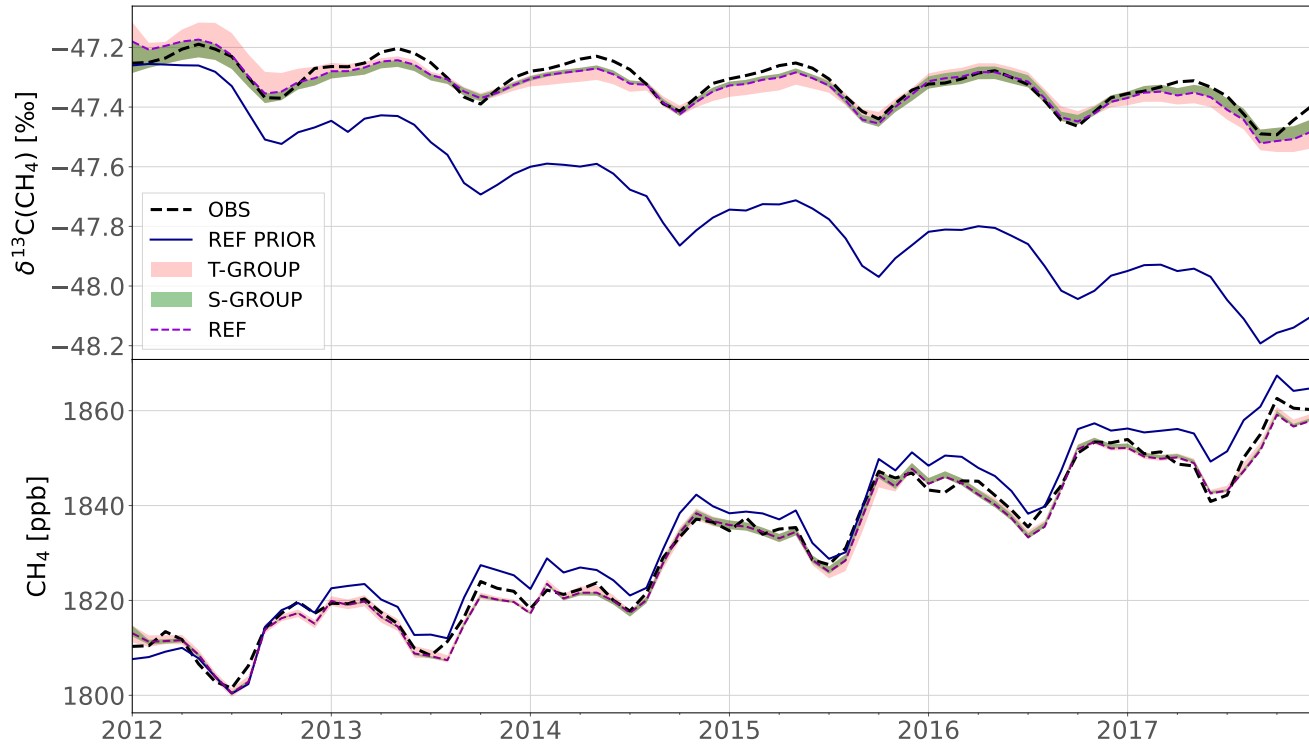

**Figure 4.** Global monthly $\delta^{13}$C(CH$_4$) and CH$_4$ means between 2012 and 2017. The dashed black and solid blue lines in each panel denote the observed and prior simulated values (REF), respectively. The red and green ranges show the maximum and minimum values of the T-group and S-group, respectively. The thick and dashed purple line denotes the posterior REF values. Globally-averaged values are computed using a method similar to Masarie and Tans (1995): a function including 3 polynomial parameters (quadratic) and 8 harmonic parameters is fitted to each time-series at available sites; the final value is obtained by performing a latitude-band weighted average over the Marine Boundary Layer (MBL) sites. The latitude band width was set at 30°. The posterior NOISO lines were not included because 1) the posterior NOISO global source signature is -54.1 ‰ and the line would therefore reach lower values than the REF PRIOR, affecting the visual clarity of the upper plot. 2) The posterior NOISO CH$_4$ values are extremely close to the REF values and including it would also affect the clarity of the lower plot.

Prior $\delta^{13}$C(CH$_4$) prescribed in REF are continuously decreasing from -47.2 to -48.2 ‰ and thus agrees very poorly (RMSE is 0.47 ‰) with observed values. This can be due to an underestimation (too negative values) of some isotopic source signatures, an underestimation of the KIE values associated with the various sinks, an underestimation of the various sinks intensities (mostly Cl and OH) and/or a poor prior estimation of the source partitioning, i.e., an underestimation of $^{13}$C-enriched sources (FF or BB) or an overestimation of $^{13}$C-depleted sources (WET or AGW). The data assimilation process reconciles simulated

and observed $\delta^{13}$C(CH$_4$) (RMSE is $0.086 \pm 0.008$ ‰) for all configurations, albeit small differences depending on the setup emerge.

The S-group provides a better match to $\delta^{13}$C(CH$_4$) observations than the T-group ($0.081 \pm 0.003$ ‰ versus $0.091 \pm 0.007$ ‰). The fit is very similar within the S-group. In contrast, the spread in the T-group is larger with $\delta^{13}$C(CH$_4$) RMSE being equal to $0.093$ ‰, $0.091$ ‰ and $0.099$ ‰ respectively for T2, T3 and T4. These results suggest that giving more freedom to the system to adjust the isotopic signatures and providing regional-specific estimates of prior source signatures instead of global values may be key elements for reaching better agreement. Best results (i.e., smallest RMSE) are obtained with T1

(0.079 ‰). However, this configuration necessitates 10 additional iterations to reach better results than REF. Without these additional iterations, REF would have the best results (0.081 ‰).

     Figure 5 shows the RMSE distribution at all measurement sites for each configuration. All sites exhibit a RMSE reduction (from prior to posterior) for both CH$_4$ and $\delta^{13}$C(CH$_4$), except for BKT with T3 and T4 configurations. Furthermore, BKT, WKT, UUM, AMY and PON exhibit a posterior CH$_4$ RMSE above 25 ppb, showing that CH$_4$ measurements retrieved at these

stations are not properly reproduced by the model, despite the optimization. Prescribed observation errors are likely not the main cause because mean values for these stations are large (10-15 ppb) but not the largest among all the assimilated stations. It can also be due to transport error or misrepresentation of sources close to the sites. Addressing this misfit is beyond the scope of this study, although the configuration influences the results : BKT and UUM fitting are notably deteriorated with T3 and T4 configurations. For example, BKT appears to be influenced by biomass burning sources in South-East Asia, which

are strongly dependent on the configuration (see Sect. 3.3). Moreover, T3 provides the poorest $\delta^{13}$C(CH$_4$) fitting at AMY (0.24 ‰). Therefore, using global values for source signatures and preventing the system from optimizing them lead to poorer fitting. On the contrary, T1 improves the results, indicating that additional degrees of freedom can help to reconcile simulations with observations, especially in South-East Asia and East Asia where these stations are located.

### 3.3    Global and regional emission increments

We are primarily interested in the additional information provided by the assimilation of $\delta^{13}$C(CH$_4$) data. Rather than discussing the regional and global CH$_4$ emissions and comparing these results to previous estimates, we investigate the differences between emissions inferred from configurations with and without the additional isotopic constraint. Long-term inversions will be run in the future with this system to provide more robust estimates of CH$_4$ emissions and compare them to the existing literature.

The inversion time-window is the 2012-2017 period. However, flux and source signature estimates in the 2012-2013 and 2016-2017 periods are not interpreted as the system appears to require a 2-year spin-up (2012-2013) and a 2-year spin-down (2016-2017), over which the inversion problem is not sufficiently constrained and isotopic signatures vary widely over time. Therefore, only the 2014-2015 estimates are analyzed in Sect. 3.3 and 3.4. Figure S2 in the supplement shows the time-series of isotopic signatures and illustrates this choice. These long effects are certainly caused by the relatively long relaxation timescales

of isotopic ratios in the atmosphere (Tans, 1997) compared to that of total CH$_4$. Fully understanding this would require a lot of time and running multiple inversions (or possibly only tangent-linear simulations), starting from different initial conditions

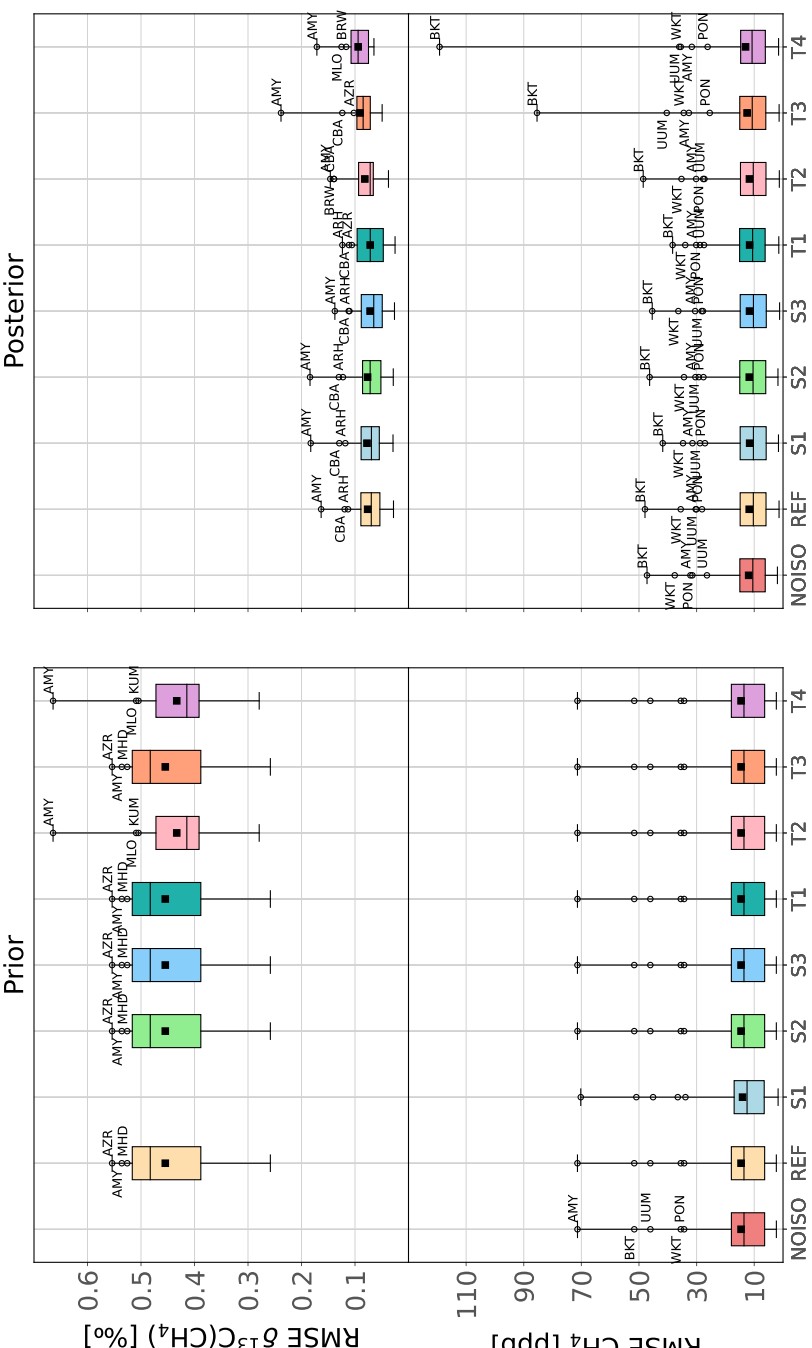

**Figure 5.** RMSE distribution over the surface stations. Left panels show the prior $CH_4$ and $\delta^{13}C(CH_4)$ RMSE and right panels show the posterior RMSE. Upper panels show $\delta^{13}C(CH_4)$ RMSE and lower panels show $CH_4$ RMSE. For clarity reasons, S1 prior is not shown for $\delta^{13}C(CH_4)$ because the associated prior misfit is much larger than that of the other configurations. The box plot whiskers are covering the whole range of data. In the lower-left panel, all station labels are identical, therefore most of them are removed to improve the clarity.

spanning the prescribed uncertainty envelope, to infer until when the initial atmospheric isotopic ratios and/or isotopic source signatures can influence the time-series of atmospheric isotopic ratios. This was too much work for this study but will certainly be addressed in future studies.

Figure 6 shows global and regional increments from the NOISO and REF inversions relative to prior estimates. Hereinafter, these differences will be referred to as "REF increment" (REF - PRIOR) and "NOISO increment" (NOISO - PRIOR). The difference between both increments will be called an "increment difference". Note that prior emissions are identical for all configurations. Posterior total emissions is $594.6 \pm 1.2$ TgCH$_4$.yr$^{-1}$ over all configurations, indicating that the isotopic constraint and setup configurations do not significantly affect posterior global emissions. A higher discrepancy between the budgets

would have indicated a malfunction in the system as the prescribed sinks are identical. The small associated standard deviation is likely caused by a slight difference in the fitting to the observations and/or by the spatial variability of the prescribed sink coupled with a small relocation of emissions depending on the configuration. For instance, OH concentrations are larger in the tropics and a relocation of emissions from the tropics to higher latitudes would be compensated for by larger global emissions. Between REF and NOISO, there is only a difference of 0.02 TgCH$_4$.yr$^{-1}$. We can therefore conclude that the additional iso-

topic constraint either relocates the emissions or reallocates them between categories, as intended. All but one of the emission categories exhibit large changes between NOISO and REF : WET, FF, AGW and BB categories.

Overall, increments are large in regions with high emissions. Global increment differences (between REF and NOISO) in AGW ($-6.4$ TgCH$_4$.yr$^{-1}$) and FF emissions ($+8.6$ TgCH$_4$.yr$^{-1}$) are mainly due to regional increment differences in China and Temperate Asia. AGW regional increment differences are equal to $-2.1$ TgCH$_4$.yr$^{-1}$ in Temperate Asia and in China.

Similarly, FF regional increment differences are equal to $+1.5$ TgCH$_4$.yr$^{-1}$ in Temperate Asia and $+5.0$ TgCH$_4$.yr$^{-1}$ in China. The WET global increment difference ($-5.7$ TgCH$_4$.yr$^{-1}$) is mainly due to differences in Canada ($-2.0$ TgCH$_4$.yr$^{-1}$) and South America ($-2.3$ TgCH$_4$.yr$^{-1}$) but other regions such as Russia, Temperate Asia and South-East Asia are involved. BB emissions are also modified when implementing the isotopic constraint. Their global increment difference is equal to $+3.2$ TgCH$_4$.yr$^{-1}$ principally owing to regional increment differences in South-East Asia ($+1.7$ TgCH$_4$.yr$^{-1}$), Canada

($+0.4$ TgCH$_4$.yr$^{-1}$) and Africa ($+0.4$ TgCH$_4$.yr$^{-1}$). The NAT category exhibit very little changes (less than 1 TgCH$_4$.yr$^{-1}$), even in relative values.

S-group configurations infer posterior results that are consistent with REF, with only small variations depending on the category and the region (see Table S5 in the supplement). In particular, S1 provides roughly the same results as REF but with more iterations, highlighting again that offsetting the initial conditions can help to reduce the computational burden without

affecting the results. On the contrary, T-group configurations are affecting the increments, although T1 and T2 configurations are generally much closer to REF than T3 and T4. T1 (yellow dot) and T2 (blue dot) exhibit differences with the S-group mainly in China where WET and FF increments are modified ($\sim -3$ TgCH$_4$.yr$^{-1}$). More importantly, almost freezing the isotopic signatures to their prior values (T3 and T4) results in increment differences 3 to 4 times larger than with REF, i.e., more than 10 TgCH$_4$.yr$^{-1}$ at the global scale. It highlights the dependence of the inferred CH$_4$ emissions to the prior source

signatures estimates. In other words, the quality of isotopic source signatures (values and uncertainties) appears to be critical for the robustness of emissions estimates.

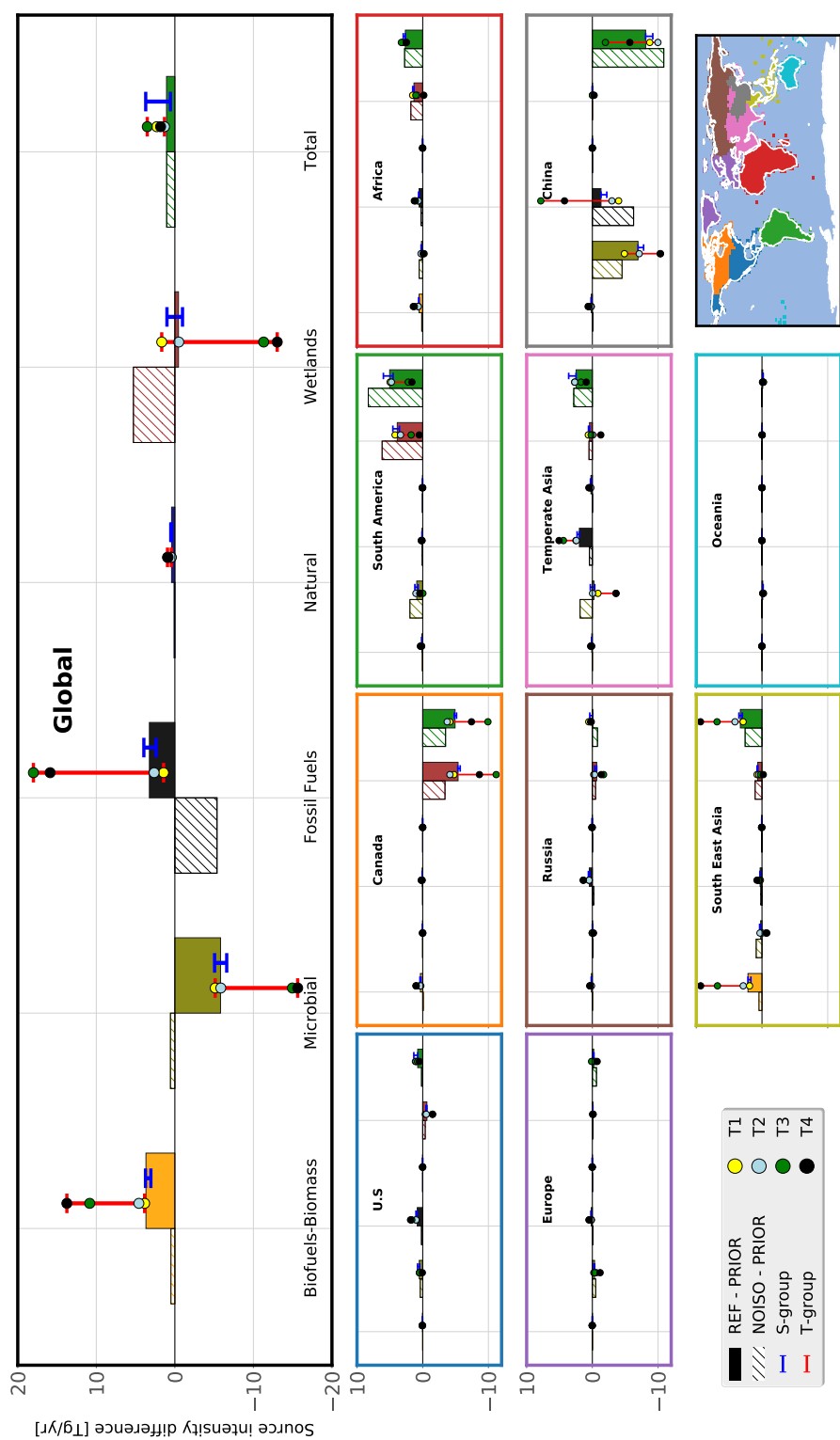

**Figure 6.** REF and NOISO emission increments for the 2014–2015 period. Prior estimates (PRIOR) are identical for both configurations. The color-filled bars show the differences between REF posterior and prior estimates (REF increment). The hatched bars show the differences between NOISO posterior and prior estimates (NOISO increment). The upper panel refers to the global emissions. The lower panels refer to multiple regions of the globe. The regions are shown on the lower right panel. Red and blue error bars represent the minimum and maximum of the T-group and S-group, respectively. Circles on the red error bar show the results from the T-group.

**Table 3.** Global $CH_4$ emissions by source category and region ($TgCH_4\,yr^{-1}$) for the REF configuration. Uncertainties are reported as the [min–max] range of all configurations.

| | BB | AGW | FF | NAT | WET | Total |
|---|---|---|---|---|---|---|
| U.S | 1 [1 - 1] | 22 [21 - 22] | 14 [13 - 15] | 2 [2 - 2] | 17 [16 - 17] | 56 [55 - 56] |
| Canada | 2 [1 - 3] | 2 [2 - 2] | 2 [2 - 2] | 1 [1 - 1] | 21 [16 - 23] | 29 [24 - 30] |
| South America | 2 [2 - 3] | 30 [29 - 31] | 6 [6 - 6] | 5 [5 - 5] | 53 [50 - 55] | 96 [93 - 99] |
| Africa | 9 [8 - 10] | 25 [25 - 26] | 14 [13 - 15] | 4 [4 - 4] | 28 [26 - 28] | 80 [80 - 80] |
| Europe | 1 [1 - 1] | 20 [19 - 20] | 6 [6 - 7] | 2 [2 - 2] | 4 [4 - 4] | 34 [33 - 34] |
| Russia | 2 [2 - 2] | 5 [5 - 5] | 12 [12 - 13] | 3 [3 - 3] | 12 [11 - 13] | 35 [34 - 36] |
| Temperate Asia | 3 [3 - 3] | 54 [51 - 56] | 28 [27 - 31] | 7 [7 - 7] | 13 [11 - 13] | 105 [104 - 106] |
| China | 5 [5 - 5] | 29 [26 - 32] | 24 [19 - 33] | 1 [1 - 1] | 5 [5 - 5] | 64 [61 - 70] |
| South East Asia | 11 [9 - 18] | 23 [22 - 23] | 8 [7 - 8] | 4 [3 - 4] | 22 [21 - 23] | 66 [66 - 72] |
| Oceania | 1 [0 - 1] | 4 [4 - 5] | 2 [2 - 2] | 1 [1 - 1] | 3 [3 - 3] | 11 [11 - 11] |
| Others | 1 [1 - 1] | 4 [4 - 4] | 5 [5 - 5] | 8 [8 - 8] | 2 [2 - 2] | 19 [19 - 19] |
| Global | 37 [33 - 47] | 220 [210 - 226] | 119 [111 - 134] | 38 [38 - 39] | 180 [167 - 185] | 594 [594 - 597] |

### 3.4 Global and regional source signature increments

Isotopic source signatures are also optimized by the system. Figure 7 provides the differences of flux-weighted source signatures between REF posterior and prior estimates for different regions and each emission category.

With configurations that allow the source signatures to be optimized, all source signatures are shifted upwards by the inversions in order to correct the excessively strong negative trend in $\delta^{13}C(CH_4)$. At the global scale, the flux-weighted source signatures of WET, FF, AGW, BB and NAT are increased by 1.7, 0.5, 0.9, 0.5 and 0.1 ‰, respectively. The global source signature is increased from $-53.9$ ‰ (prior) to $-52.6 \pm 0.2$ ‰ (posterior with standard deviation over the configurations). More information is provided in the supplement (Table S6). The posterior global signature is strongly dependent on the KIE of atmospheric oxidation. This effect tends to deplete air in $^{13}CH_4$, shifting the $\delta^{13}C(CH_4)$ to more positive values as the $CH_4$ molecules emitted by the sources are removed from the atmosphere. The mean KIE in our simulations depends on 1) the prescribed OH, $O(^1D)$ and Cl concentrations and 2) the prescribed KIE values associated to the individual sinks. As the mean KIE is the same for all configurations, the posterior global source signatures are very close.

The WET global source signature, associated with REF posterior estimates, exhibits the larger upward shift compared to prior estimates, from a value of $-60.8$ ‰ to $-59.1$ ‰. Large upward WET source signature shifts are located in boreal regions (North America, Russia) but also in South America and Temperate Asia. The AGW source signature is increased by 0.9 ‰ mainly due to changes in Asia. The FF source signature is increased by 0.5 ‰ due to a large increment in China ($+1.2$ ‰). Finally, the BB source signature is modified in South-East Asia ($+1.4$ ‰) and Canada ($+0.8$ ‰).

These changes are consistent within the S-group (see blue errorbars in Fig. 7), although small variations are visible (e.g., $\pm\,0.3$ ‰ for WET in Canada). The source signature is therefore modified nearly to the same extent in all regions, no matter which configuration in the S-group is analyzed. T1 (see yellow dot in Fig. 7), with more optimized categories than the others,

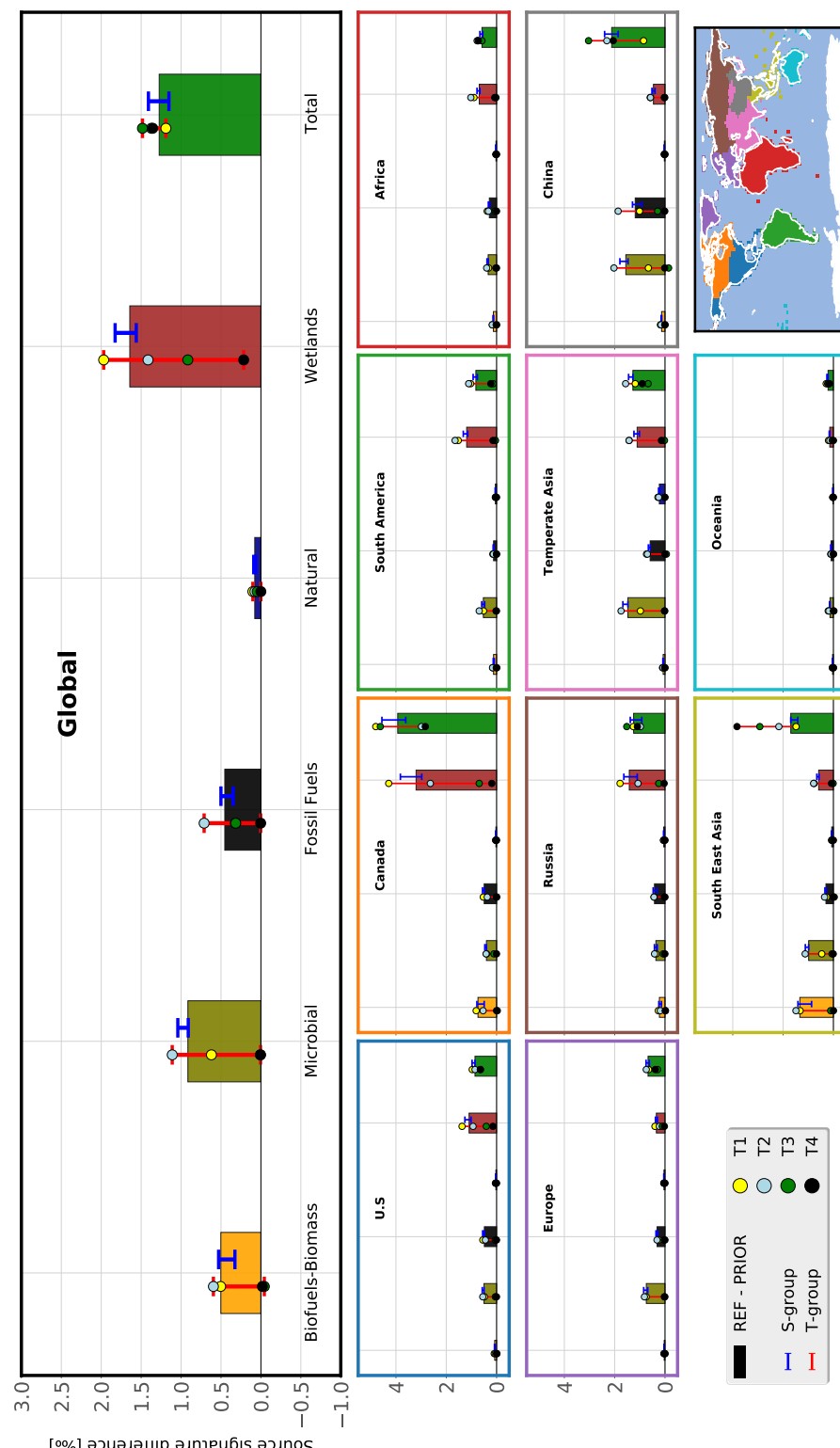

**Figure 7.** REF flux-weighted source signature increments for the 2014-2015 period. The color-filled bars show the differences between REF posterior and prior estimates (REF increment). The upper panel refers to multiple regions of the globe. The regions are shown on the lower right panel. The lower panel refers to the global emissions. Red and blue error bars represent the minimum and maximum of the T-group and S-group, respectively. Circles on error bars show the results from the T-group.

shows small differences at the global scale (less than 0.3 ‰ for all categories), although differences of more than 1 ‰ are found in China. Therefore, increasing the number of degrees of freedom lead to similar flux estimates but can affect the signatures at regional scale. T2 estimates are shifted upward to reach a less negative global isotopic source signature without getting closer to the regional distribution of the S-group. This is likely caused by the scarcity of $\delta^{13}C(CH_4)$ stations and correcting this behavior seems challenging without additional observations. The problem might be circumvented by using the region scale rather than the pixel scale to optimize isotopic signature values. Future inversions will test this assumption.

These results must be interpreted with caution because the input data suffer from high uncertainties. The artificial increase of the source signatures by our system can be hardly related to litterature and former investigations. Consequently, it is challenging to conclude whether an increase of the source signatures would be more realistic (i.e., supported by observational data) than, for instance, only increasing the emissions of $^{13}C$-enriched sources such as BB. This system is only based on a mathematical and physical framework connecting the several groups of uncertainties (observational, prior fluxes, prior source signatures, prior sinks) and finding the most likely solution. Better estimates of these uncertainties must be prescribed before obtaining robust results. In particular, the uncertainties on KIE values and sink intensities have not been tested here and could largely influence the results. Also, the uncertainties on source signatures are relatively smooth in REF compared to recent country-specific estimates (Sherwood et al., 2017). Assessing these uncertainties should be a key aspect for future studies using this new inversion system to quantify the global $CH_4$ budget.

### 3.5 Posterior uncertainties

Formally, posterior uncertainties are given by the Hessian of the cost function. This matrix can hardly be computed at an achievable cost considering the size of the inverse problem. Other means must be implemented to get posterior uncertainty such as estimating lower-rank approximation of the Hessian, using Monte-Carlo ensembles of variational inversion to represent the prior uncertainties or computing multiple configurations covering a given range of possibilities. Here, using multiple configurations provides insight into the posterior uncertainty associated with the posterior fluxes. We calculated the full uncertainty range using the minimum and maximum values among all the configurations, as in Saunois et al. (2020). WET, AGW, FF and BB flux estimates (Table 3) exhibit an uncertainty of 10 %, 7 %, 19 % and 38 %, respectively. BB is the most uncertain estimate relative to its intensity, although FF shows the largest absolute uncertainty (23 $TgCH_4.yr^{-1}$). These uncertainties are unlikely to be affected by the assimilation of additional $\delta^{13}C(CH_4)$ data because we expect the uncertainties on the isotopic source signatures to have a much larger influence. However, this remains to be tested in future work if posterior uncertainties can be calculated.

At present, M1QN3 is not the only optimization algorithm that can be utilized to perform variational inversions in the CIF. The CONGRAD algorithm (Fisher, 1998), that follows a conjugate gradient method combined with a Lanczos algorithm, is also implemented. In particular, it considerably facilitates the computation of posterior uncertainties. Any change in algorithm is very easy and accessible to any CTM embedded in the CIF. However, CONGRAD has not been tested yet with $\delta^{13}C(CH_4)$ data. As CONGRAD is only designed for linear problems, using this algorithm could radically change the results of inversions

performed with the isotopic constraints and future work will focus on using CONGRAD to perform the inversions with isotopic constraints.

## 4   Conclusions and perspectives

We present here a new variational inversion system designed to assimilate observations of both a specific trace gas and its isotopic data. This system allows to optimize both tracer emissions and associated isotopic signatures for multiple source

categories. To test this system we have assimilated $CH_4$ and $\delta^{13}C(CH_4)$ data retrieved at different measurement sites over the globe.

Different configurations have been tested in order to assess the sensitivity of the system to the setup. We have shown that offsetting the $\delta^{13}C(CH_4)$ initial conditions before the inversion (S1), using $\delta^{13}C(CH_4)$ curve fitting data instead of the original observations (S2) and reducing the prescribed uncertainties in the $\delta^{13}C(CH_4)$ observations (S3) have very little effect

on the inferred fluxes (less than 2 $TgCH_4.yr^{-1}$ for each category at the global scale). However, offsetting $\delta^{13}C(CH_4)$ initial conditions before the inversion results in a reduced computational time (21 less iterations).

Other setup choices have more influence on the results. Increasing the number of source categories (T1) requires more computational time (10 more iterations) to reach a cost function (and RMSE) reduction similar to REF. Moreover, although the global posterior emissions with an increased number of categories are very close to those inferred with REF (less than

1 $TgCH_4.yr^{-1}$), the posterior isotopic signatures can be modified in some regions (more than 1 ‰ in China). Also, starting from globally-averaged values for the source signatures (T2) makes the system unable to retrieve the regional-specific isotopic signatures from REF. Increasing the number of $\delta^{13}C(CH_4)$ observations could help to cope with this issue. Finally, configurations constraining the source signatures (T3-T4) show differences in global flux estimates of more than 10 $TgCH_4.yr^{-1}$, compared to REF. This emphasizes the need for good prior source signature estimates.

The major drawback of this inversion system is undoubtedly the large computational burden of a full minimization process. At least 40 iterations appear to be necessary to reach a satisfying convergence state at the regional scale. For the LMDz-SACS model, a maximum of 8 CPUs can be run in parallel, resulting in an elapsed time of 5-6 weeks to run one of the inversions of this study. A new generation of transport models such as DYNAMICO (Dubos et al., 2015) could help to address this problem in the future by allowing more processors to run in parallel. Also, further developments will implement some parallelization

methods to enable computational burden reduction (e.g., Chevallier, 2013). In addition, variational inversions as implemented in the CIF do not provide a quantification (even approximated) of the posterior uncertainties. Dedicated efforts need to be done to address this issue in the future, at an achievable numerical cost. In particular, using the CONGRAD algorithm instead of M1QN3 could be a solution as both algorithms can be easily selected in the CIF. However, additional work is needed to ensure that switching the optimization algorithm does not affect the results inferred with our new system.

This system is implemented within the CIF framework and can therefore be used for inversions with the various CTMs embedded in the CIF, provided the adjoint codes of the models exist. As the operations developed for the purpose of this study are performed outside the model structure, forward, tangent-linear and adjoint codes from other CTMs do not require any

modifications as long as the model is capable of simulating both $^{12}CH_4$ and $^{13}CH_4$ simultaneously. The prior input must be adapted to the new model (spatial and time resolution) but the format of the observational data and of the prescribed errors can be preserved. Also, due to the variational method benefits, the efforts dedicated to the preparation of inputs do not scale with either the size of the observational datasets or the length of the simulation time-window. Therefore, this system is very powerful and is particularly relevant to study in a consistent way the influence of multiple physical parameters on atmospheric isotopic ratios, such as the transport, the isotopic signatures, the emission scenarios, the KIE values, etc. We did not try to assess here the sensitivity of the system to these parameters as only technical aspects of the system were tested. This will be part of future analyses.

As mentioned in the introduction, future work will address the estimation of $CH_4$ emissions over longer periods of time using this new system. For instance, the 2000-2006 $CH_4$ stabilization period and the subsequent renewed growth are particularly interesting to study using the isotopic constraint as global $\delta^{13}C(CH_4)$ started to decrease after 2006. These periods of time have already attracted considerable critical attention from many inversion studies (with or without the isotopic constraint) and comparing the results derived from such a complete 3-D variational inversion system with other recent estimates should be highly relevant. The most important limitation of assimilating $\delta^{13}C(CH_4)$ lies in the fact that very limited $\delta^{13}C(CH_4)$ data are available, and therefore evaluating the posterior simulated $\delta^{13}C(CH_4)$ is often challenging, if not impossible. However, satellite and balloon / AirCore data can easily be used to evaluate the posterior simulated $CH_4$.

$\delta^{13}C(CH_4)$ is not the only isotopic data that can be assimilated in such a system. Many $\delta D(CH_4)$ observations have also been retrieved during the 2004-2010 period at many different locations. These isotopic values can provide additional information that can further help to discriminate the co-emitted $CH_4$ fluxes (Rigby et al., 2012). Moreover, ethane ($C_2H_6$) is co-emitted with $CH_4$ by fossil fuel extraction and distribution (Kort et al., 2016; Smith et al., 2015) and observations are available at a multitude of sites since the early 1980s. Therefore, assimilating this data can provide additional constraint. The system will therefore be improved in the future in order to assimilate $\delta^{13}C(CH_4)$, $\delta D(CH_4)$ and $C_2H_6$ observations together.

*Data availability.* The code files of the CIF version used in the present paper are registered under the following link : https://doi.org/10.5281/zenodo.6304912. Prior anthropogenic fluxes (EDGARv4.3.2) can be downloaded from the EDGAR website (https://edgar.jrc.ec.europa.eu/dataset_ghg432). Biomass burning fluxes can be downloaded from the GFED website (https://globalfiredata.org/pages/data/). Prior natural fluxes and other data are available upon request (joel.thanwerdas@lsce.ipsl.fr). The $CH_4$ data used in the present paper were provided by many stations and measurement networks around the world (a comprehensive list can be found in the supplement and in the acknowledgments). Their data is freely available upon request to the station maintainers or via dedicated websites. The $\delta^{13}C(CH_4)$ observational data can be downloaded from the NOAA-GML website (https://gml.noaa.gov/dv/data/).

*Author contributions.* JT implemented the variational inversion system within the CIF with the precious help of AB. JT designed, run and analyzed the tested configurations. AB, MS, IP and PB provided scientific and technical expertise. They also contributed to the analysis of

this work. BV and SEM provided the $\delta^{13}C(CH_4)$ data and scientific expertise regarding $\delta^{13}C(CH_4)$ observations. JT prepared the manuscript with contributions from all co-authors.

*Competing interests.* The authors declare that they have no conflict of interest.

*Acknowledgements.* This work was supported by the CEA (Commissariat à l'Energie Atomique et aux Energies Alternatives). The study extensively relies on the meteorological data provided by the ECMWF. Calculations were performed using the computing resources of LSCE, maintained by François Marabelle and the LSCE IT team. The authors thank the reviewers (Peter Rayner and two anonymous referees) for their fruitful comments and suggestions on our manuscript. We are also grateful to many station maintainers that provided the $CH_4$ data: **For AGAGE network** - AGAGE is supported principally by NASA (USA) grants to MIT and SIO, and also by: BEIS (UK) and NOAA (USA) grants to Bristol University; CSIRO and BoM (Australia): FOEN grants to Empa (Switzerland); NILU (Norway); SNU (Korea); CMA (China); NIES (Japan); and Urbino University (Italy); **For South African Weather Service Cape Point GAW station** - We thank Casper Labuschagne, Thumeka Mkololo and Warren Joubert ; **For EC network** - We thank Doug Worthy; **For MGO network** - We thank Nina Paramonova and Victor Ivakhov; **For AMY network** - We thank Haeyoung Lee and Sepyo Lee - The AMY data was funded by the Korea Meteorological Administration Research and Development Program under grant KMA2018-00522; **For ENEA network** - We thank Alcide Disarra, Salvatore Piacentino and Damiano Sferlazzo; **For EMPA network** - We thank Martin Steinbacher and Brigitte Buchmann - The $CH_4$ measurements at JFJ are run by EMPA in collaboration with FOEN and are also supported by ICOS-CH; **For BMKG-EMPA network** - We thank Martin Steinbacher, Alberth Christian and Sugeng Nugroho - The $CH_4$ measurements at BKT are run by BMKG in collaboration with EMPA;

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
