# Peer review of "Variational inverse modeling within the Community Inversion Framework v1.1 to assimilate $\delta^{13}C(CH_4)$ and $CH_4$ : a case study with model LMDz-SACS"

_Geoscientific Model Development, 2021_

## Referee Comment (RC3)

Review on Thanwerdas et al., gmd-2021-106

**General comments**
The paper presents development of an atmospheric inversion model of methane ($CH_4$) by assimilating both $CH_4$ and $\delta^{13}C(CH_4)$ observations in order to optimizing global and regional $CH_4$ fluxes. The method allow to optimize several source sectors simultaneously, which is an advantage over $CH_4$ only assimilation methods that is more suitable for optimizing total budgets. As $CH_4$ is an important greenhouse gas, which has high mitigation potential, it is urgent to understand current budgets of both anthropogenic and natural sources. This work is highly relevant and valuable for increasing understanding regional source-specific emissions. Furthermore, the method is developed for Community Inversion Framework (CIF), which is flexible in various inverse modelling methods, such as transport models and optimization method. Therefore, this development will be beneficial for all CIF users.

The manuscript is generally well written and presented. I recommend the manuscript to be published, but would like to point out a few comments which could increase the value of the paper.

Presentation of novelty
As authors mention very briefly, this is a first attempt to carry out variational inversion assimilating $\delta^{13}C(CH_4)$ observations. Please mention this also in the abstract, and add slightly more details of the development in the Introduction, e.g. development of adjoint and implementation in CIF. From the Introduction, I was also not sure if such modelling has been done with LMDz previously, i.e. how well LMDz have been simulating $\delta^{13}C(CH_4)$?

Categorization of the simulations
I was not completely convinced about those S and T groups. Are they really needed? Did you categorize them based on results or really expected T groups to have higher variation before you started simulations? T1 is not only about changing isotope signature values and their uncertainty, but also degree of freedom (dof) in the optimization (I guess you optimize 10 flux categories?).

Discussion on results
Although this is a technical paper is not meant to evaluate the flux estimate nor $\delta^{13}C(CH_4)$ values obtained from the simulations, I would like to see briefly how your estimates are compared to previous studies. Or even simply mentioning in the Conclusion how you would do further analysis, including e.g. availability of evaluation data.

Discussion on uncertainty estimates
I understand that it is costly to calculate the full uncertainty from all simulations. However, you anyway present uncertainty in P12 L12. How was it calculated? From the cost function, you can speculate how dof and inclusion of additional data would affect the posterior uncertainty. Please comment on it in Section 3.5.

**Specific comments**
Method
Distribution of state vectors: did you assume all to be normal/Gaussian?

How did you derive the aggregated signature values?

What is the temporal and spatial resolution of prior fluxes?

What is the temporal resolution of the optimization?

Can you provide range of observation uncertainty (diagonals of **R**) for each stations, maybe by adding information in Table S3 and S4, and briefly mention ranges in the main text? This will help understanding the results on cost function and RMSE differences better.

Curve fitting data:
- Was there any specific reason why you decided to use smoothed data?
- After curve fitting, what is the temporal resolution of the data you assimilated? Did you generate same amount of $\delta^{13}C(CH_4)$ data in REF and S2?

Offsets in initial condition: How much offset did you need to add/subtract?

Results
P13 L9: "Consequently, the system is preferentially adjusting $\delta^{13}C(CH_4)$ over $CH_4$ values to reduce the cost function."
- Can you speculate why? Is it because observation uncertainty (diagonals of **R**) is relatively smaller in $\delta^{13}C(CH_4)$ than $CH_4$? The cost function show that the observational constraint in $CH_4$ is larger (probably main reason is amount of data?).
- For S2, I wonder why contributions of $\delta^{13}C(CH_4)$ and $CH_4$ are similar to S3. Did you assimilate same amount of $\delta^{13}C(CH_4)$ data in REF and S2?

P14 L30-34:
This could also be due to prescribed observation/transport model uncertainty.

Emission increments: Emission changes are large in regions with high emissions. Please mention.

Conclusion
Please expand how much work would be needed for switching transport models and optimization methods in CIF for the $\delta^{13}C(CH_4)$ data assimilation. Can we use e.g. initial mixing ratios, do we need to run spin-up and build adjoint if transport model is changed? How about changes in optimization methods? Can we use same state vectors and covariance structures?

Figures
Figure 3: Please add label of x-axis

Figure 4: Please add legend of posterior results from REF simulation, and perhaps use different color than green, as it's not S-group simulation? Please also add results from NOISO.
Figure 4 caption: I guess the figure is global **monthly** mean?

Figure 5: Prior $CH_4$ is same for all simulations, and $\delta^{13}C(CH_4)$ for some. Please consider minimizing.

Figure S2: Please add label and unit of y-axis. Caption is slightly unclear – what do you mean by "inferred with REF"?

**Technical comments**
P14 L22: The S-group provides a better match **to δ13C(CH4) observations** than...

P15 L4-5: AMY is not in South-East Asia.

---

## Author Response (AR1)

We are very grateful to the three referees for their detailed and fruitful comments which have allowed us to clarify various points. We copy-pasted below their reviews. Comments from Reviewers #1, #2 and #3 are in red, green and blue, respectively. For each comment/suggestion, our responses are in bold black and the revised/additional text in italic black. We also provide a track-change manuscript at the end of the present document.

Also, please note that we gathered all the questions/comments about the curve fitting (pseudo-observations) in the same paragraph (pages 5-6)

**General and specific comments**

**Reviewer #1**

Information about data and code availability is lacking. Under "Data availability" the authors give http://community-inversion.eu as the reference for the code, however, this is just a general website about the CIF. This website indicates the Git git.nilu.no/VERIFY/CIF but this is just for the generic version of CIF and not that pertaining to this paper. Also, under this section, details about where to access the observational and prior flux data should be given.

We included a DOI for this version of the CIF and additional information about where to access the observational and prior flux data.

The code files of the CIF version used in the present paper are registered under the following link : https://doi.org/10.5281/zenodo.6304912. Prior anthropogenic fluxes (EDGARv4.3.2) can be downloaded from the EDGAR website (https://edgar.jrc.ec.europa.eu/dataset\_ghg432). Biomass burning fluxes can be downloaded from the GFED website (https://globalfiredata.org/pages/data/). Prior natural fluxes and other data are available upon request (joel.thanwerdas@lsce.ipsl.fr). Many stations from different networks contributed to the CH4 data used in the present paper (a comprehensive list can be found in the supplement and in the acknowledgments). Their data is freely available upon request to the station maintainers or via dedicated websites. The  $\delta^{13}C(CH_4)$  observational data can be downloaded from the NOAA-GML website (https://gml.noaa.gov/dv/data/).

P1L2: suggest changing this to: "...indicating relative changes in the sources and sinks" as it is evident from the fact that the mixing ratios have been increasing that there must be a change in the sources and/or sinks and not just a variation but a change in one relative to the other.

**We agree and, following your suggestion, this part of the sentence has been modified.**

P2L13: I think this sentence is potentially confusing and could be better formulated. What I think the authors mean is that without regularization the inverse problem is ill-conditioned (or ill-posed) giving no unique solution, hence the need for regularization e.g. by providing prior information. Also it is unclear to me what is meant by "no continuity with the data" - could the authors please explain this.

We apologize for not making this sentence clear enough. An ill-posed problem is often defined as a problem which may have more than one solution but also in which the solutions depend discontinuously upon the initial data. In an inversion problem, the data referred to the atmospheric observations and we know that a small change in this data could result in a radically different solution. We used your suggestion to modify this sentence and make it more intelligible.

Without regularization of the problem, e.g. providing prior information, the inverse problem is ill-conditioned (or ill-posed). It means that there is no unique solution to the problem but also that a small error in the assimilated data (here atmospheric observations) can result in large errors in the derived solution.

P2L21: Variational methods, such as the Lanczos version of the conjugate gradient algorithm provides the posterior error covariance matrix with little additional computational cost.

We agree with the reviewer. We should have added that we consider our inversion problem (our observation operator) as non-linear because we include isotopic observations in our data. The Lanczos version of the conjugate gradient algorithm can be utilized only if the observation operator is linear. We also included a little discussion on this in Sect. 3.5 and in the conclusions because using the Lanczos method to provide posterior uncertainties is an important perspective of our work. However, it necessitates further developments and to clarify whether our observation operator and therefore our cost function can be linearized without affecting our results.

Thus, the variational formulation is preferred to the others when optimizing emissions and sinks at the pixel scale using large volumes of observational data, although its main limitation is the numerical cost to access posterior uncertainties when there is non-linearity in the inversion problem (Berchet et al, 2021).

P2L33-34: I would suggest the authors give ranges for the various source categories to reflect how variable values within each category can be.

We agree with this suggestion. Although we preferred not to mention the minimum and the maximum provided by Sherwood et al. (2017) because using isotopic data would appear less relevant to the reader. We preferred to include the mean and standard deviations for each source process.

CH4 isotopic source signatures  $\delta^{13}C(CH_4)_{source}$  notably differ between emission categories ranging from 13C-depleted biogenic sources (-61.7 ± 6.2 ‰, one standard deviation) and thermogenic sources (-44.8 ± 10.7 ‰) to 13C-enriched thermogenic sources (-26.2 ± 4.8 ‰) (Sherwood et al., 2017; Schwietzke et al., 2016), although the distributions are very large and overlaps exist between the extreme values.

P3L2: I think the authors should precise that they are not consistent with the d13C observations and the prescribed d13C ratios.

**We added a sentence to elaborate.**

Saunois et al. (2017) pointed out that many emission scenarios inferred from atmospheric inversions are not consistent with  $\delta^{13}C(CH_4)$  observations and that this constraint must be integrated into the inversion systems to avoid such inconsistencies. In addition, they highlighted the sensitivity of the atmospheric isotopic signal to the source partitioning and prescribed isotopic ratios.

P3L12: Thompson et al. 2018 used a variational method to optimize CH4 emissions and the OH sink with the AGAGE 12-box model. Perhaps the authors mean never in a variational inversion framework with a full 3D atmospheric transport model?

Yes, we apologize because we were not clear enough and writing that, we unintentionally diminished the work of Thompson et al. (2018). We modified this sentence following your suggestion.

The implementation of such a constraint in an inversion system have already been attempted in previous studies focusing on CH4 (e.g., Thompson et al., 2018; McNorton et al., 2018; Rigby et al., 2017; Rice et al., 2016; Schaeferet al., 2016; Schwietzke et al., 2016; Rigby et al., 2012; Neef et al., 2010; Bousquet et al.,

2006; Fletcher et al., 2004) but, to our knowledge, never in a variational system associated to a 3-D chemistry-transport model (CTM).

P4L15: All Bayesian methods require the inverses of R and B.

**We agree and modified this sentence.**

As in analytical and ensemble methods, the variational formulation necessitates the inversion of both error matrices R and B.

P4L17: I think you should specify the assumption, i.e. that the observation errors are uncorrelated.

**We agree and modified this sentence.**

*R* is considered diagonal as point observations are distant in time and space (i.e., uncorrelated observation errors), allowing for the inverse to be calculated easily, although that assumption should be revised with the increasing availability of satellite sources.

EQ6-7: I'm confused about the value MTOT, is this the molar mass of CH4 in source FiTOT, if so then MTOT depends on the d13C ratio of CH4 in FiTOT.

We apologize for not mentioning this in the submitted manuscript. In our study, we set  $M_{TOT}$  ( $M_T$  in the revised manuscript), the total CH4 molar mass, to a constant value of 16.0415. As of now, this value can be freely defined by the user before starting the inversion but remain constant throughout the minimization. Molar masses are involved only to convert CH4 mass fluxes into 12CH4 and 13CH4 mass fluxes in the forward and tangent-linear runs and also to perform the equivalent operation in the adjoint run. We demonstrate very quickly here that the  $\delta^{13}C(CH_4)_{source}$  range of values observed in the CH4 sources would result in a negligible variation around the chosen value and would very likely not affect the results of our study or that of any other inversion performed with our system.

To make the demonstration, we take the  ${}^{12}CH_4$  and  ${}^{13}CH_4$  molar masses provided by Stolper et al. (2014)

M12 = 16.031 g/mol M13 = 17.035 g/mol

By definition of the total CH4 molar mass, here denoted MT, we have :

$$M_T = \frac{M_{12} + A \cdot M_{13}}{1 + A}$$

with

$$A = (1 + \delta^{13} \mathrm{C}(\mathrm{CH4})) \cdot R_{\mathrm{std}}$$

In this case,  $\delta^{13}C(CH_4)_{source}$  can roughly vary between -70 ‰ and -10 ‰. It gives  $M_T = 16.041384$  and  $M_T = 16.042046$ , respectively, thus a variation of 0.004 %. In Eq. 5 and 6 in the manuscript, both right-hand members are divided by  $M_T$ , hence the same value is applied to both fluxes. As ratios between  ${}^{12}CH_4$  and  ${}^{13}CH_4$  quantities are more important than the  ${}^{12}CH_4$  and  ${}^{13}CH_4$  values, we expect the impact of setting the total CH4 molar mass constant to be highly negligible.

Deriving tangent-linear and adjoint operations while including the relationship above would result in an overly complex code and would very likely not influence the results.

We added some explanations to the manuscript.

 $M_{T}$  should preferably depend on  $M_{12}$  and  $M_{13}$  when converting the mass fluxes:

$$M_T = \frac{M_{12} + A^i \cdot M_{13}}{1 + A^i}$$

However, the complexity of the forward, tangent-linear and adjoint codes would be largely enhanced by such a relationship. The code structure would also be less generic, i.e., it could not be used for other isotopologues of  $CH_4$ , such as  $\delta D(CH_4)$ . We choose to implement  $M_{\tau}$  as a constant that can be prescribed freely by the user, therefore without considering any influence of the prescribed  $M_{12}$  and  $M_{13}$  values, also prescribed by the user. As the observed isotopic source signatures roughly vary between -70 ‰ and -10 ‰, a maximum variation of 0.004 % in  $M_{\tau}$  could be expected. It will very likely not affect the results of our study or that of any other inversion performed with our system.

P7L13: For the category "fossil fuels" could the authors please specify if this is only fugitive emissions or also combustion emissions, and if the source signature is considered the same for fugitive and combustive emissions?

As mentioned in the main manuscript, we adopted the prior CH4 emissions compiled for inversions performed as part of the Global Methane Budget (Saunois et al., 2020). The Table below shows the relationship between our subcategories and those of EDGARv4.3.2.

The EDGARv4.3.2 categories PRO\_OIL and PRO\_GAS (fugitive emissions during oil and gas exploitation) largely contribute (~90 %) to the total of the "Oil, Gas & Industry" subcategory. Therefore, we chose to ignore the influence of other subcategories on the isotopic signature of the category. We added this sentence to the text :

The EDGARv4.3.2 categories PRO\_OIL and PRO\_GAS (fugitive emissions during oil and gas exploitation) largely contribute (~90 %) to the total of the "Oil, Gas & Industry" sub-category. Therefore, we chose to neglect the influence of other subsub-categories (such as industry) on the isotopic signature of the category.

| Grouped
sector      | EDGAR
categories  | Description                                                          | Propagation                              | IPCC categories                                                           |  |
|------------------------|----------------------|----------------------------------------------------------------------|------------------------------------------|---------------------------------------------------------------------------|--|
| Coal                   | HDC                  | hard coal                                                            | Combined. Then scaled by BP.             | 1B1a                                                                      |  |
|                        | OIL                  | Oil exploration, production, transportation,
refining and storage | Scaled by BP.                            | 1B2a1 + 1B2a2 + 1B2a3 + 1B2a4                                             |  |
|                        | GAS                  | Natural gas venting and flaring                                      | Scaled by BP.                            | 1B2c                                                                      |  |
|                        | PRO –
(OIL + GAS) | Oil evaporation from trucks and tankers +
gas from pipelines.     | This is not scaled. It is kept constant. | ?                                                                         |  |
| Oil, Gas &
industry | TNR (All)            | Transport ex road – All Aviation, Shipping,
Railways etc.         | Linearly propagated to 2017.             | 1A3a_CDS + 1A3a_CRS + 1A3aLTO
+ 1A3a_SPS + 1A3c + 1A3e + 1A3d +
1C2 |  |
|                        | TRO                  | Road transport                                                       | Linearly propagated to 2017.             | 1A3b                                                                      |  |
|                        | CHE                  | Chemical processes                                                   | Combined. Then linearly propagated to    | 28                                                                        |  |
|                        | IRO                  | Iron and steel production                                            | 2017.                                    | 2C1c + 2C1d + 2C1e + 2C1f + 2C2                                           |  |
|                        | ENE                  | Power industry                                                       |                                          | lAla                                                                      |  |
|                        | IND                  | Combustion for manufacturing                                         | 2017.                                    | 1A2                                                                       |  |
|                        | REF_TRF              | Oil refineries and transformation industry                           |                                          | 1A1b+1A1c+1A5b1+1B1b+1B2a5+1B
2a6+1B2b5+2C1b                           |  |

**Combining sectors**

P9L2: I think it would be good to include the references for the source signatures in the main part of the manuscript and not just in the supplement. Also, there is no reference given for the livestock category nor an explanation why this category had a time varying source signature and what the dependence on time was.

The references for the source signatures have been included in the main part of the manuscript (Table 1). The livestock reference was in the main part of the manuscript (P7L28). We added some explanation about the time varying component and reworded the sentences.

Livestock isotopic source signatures are taken from Chang et al. (2019) and aggregated into the 11-regions map by selecting region-specific values. Livestock source signatures have been likely decreasing over time since the 1990s due to changes in C3/C4 diet within the major livestock producing countries and therefore annual values are prescribed. However, these estimates end in 2013 and we set the years 2014 to 2017 equal to the year 2013. Consequently, only the year 2012 has a different prescribed value from the other years.

P10L7: Do the authors mean that the model, LMDZ-SACS cannot reproduce the high temporal frequency of CH4 or d13C or both? If it is d13C, weekly observations are not high frequency. Also do the authors have an idea why the temporal variability could not be reproduced? I think this needs to be better explained. Also why assimilating the curve fitted data was chosen as the solution rather than e.g. increasing the observation uncertainty, filtering or averaging the observations?

Curve fitting data:

• Was there any specific reason why you decided to use smoothed data?

• After curve fitting, what is the temporal resolution of the data you assimilated? Did you generate same amount of  $\delta$  13 C(CH4) data in REF and S2?

**Using Smoothed Observations**

I note the comment on Page 10. "The observed high- frequency temporal variability cannot be adequately reproduced by the LMDz-SACS model. Therefore, instead of assimilating the real observations, we used a smooth curve fitting the real observations." This is both striking and concerning. We noted from the earliest days of using high-frequency observations in formal inversions (Law et al., 2002, 2003; Peylin et al., 2005) that much of the power of high-frequency measurements came from the interplay between variations in meteorology and concentration. Abandoning this deserves more comment. What evidence do you have of the failure of LMDZ-SACS to simulate such observations? If you are using smoothed concentrations do you smooth the meteorology or the simulated concentration and (potentially) sensitivity the same way?

We apologize for not making this part of the explanation clearer. LMDz is obviously capable of reproducing high-frequency temporal variability. However, in this study, we focused on constraining monthly and annual flux variations (i.e. long-term trend and seasonal cycle) rather than investigating daily or weekly variations. Therefore, when assimilating isotopic observations, we quickly noticed that the monthly and annual components of the isotopic time-series had much more impact on the results than weekly (potentially large) variations in the observations.

Using the curve fitted data is equivalent to taking all observations, but with correlations in the R matrix. But as we do not want to invert a R matrix that is non-diagonal, we prefer to use pseudo-observations, filtering out the high-frequency signal.

Also, before running the inversions, we thought that maybe, using the curve fitted data would reduce the computational burden of the inversion and facilitate the convergence. However, considering S2 results, we were wrong. We chose to curve fitted the data because it appeared to be the best way to preserve the long-term trend as well as the seasonal cycle in the most intelligent way, following Masarie and Tans (1995). We sampled the pseudo-observations at the same time as the real observations.

**When referring to observations that have a "high temporal frequency", we meant observations that are not monthly or yearly averages. The term was misleading and we changed that.**

In this study, we focused on estimating monthly and annual flux variations rather than investigating daily or weekly variations. Prescribing error correlations in the R matrix (introduced in Sect. 2.1) can be used to ensure that the inversion preferentially constrains the components we are interested in (i.e., long-term trend and seasonal cycle). In order to keep the R matrix diagonal and to focus on monthly and annual variations of the signal, we chose to use  $\delta^{13}C(CH_4)$  observational data based on a curve fitting the original  $\delta^{13}C(CH_4)$  observations. The fitting curve is a function including 3 polynomial parameters (quadratic) and 8 harmonic parameters as in Masarie and Tans (1995). After the fitting, the pseudo-observations were sampled at the same time as the original observations. We also hypothesized that the convergence would be slightly faster if a smooth curve fitting the real observations was used instead of the real observations, which appeared to be false (see Sect. 3.1). One sensitivity inversion aims at estimating the error introduced by this simplification (simulation S2 in Table 2).

Fig. 3a) I think here "cost" (or "value of cost function") is meant and not "cost function" and it would help to specify that the x-axis is "iterations".

**Following your suggestion, we modified this.**

Section 3.1: I think somewhere the results of the adjoint tests should be presented since a new version of the model was developed, including its adjoint.

**In the main text :**

In order to confirm that the several adjoint operations have been correctly implemented, we also provide the results of multiple adjoint tests in the supplement (Text S4).

**In the supplement (Text S4) :**

The adjoint code test is based on the definition of the adjoint observation operator :

$$\langle H\delta U, H\delta U \rangle = \langle \delta U, H^*H\delta U \rangle$$

In practice, the vector  $\delta U = \lambda \cdot U$  is first provided as an input to the tangent-linear model, with  $\lambda$  being a scalar. After this, the output vector  $\delta U$  is retrieved and the first scalar product (left-hand member) is calculated. The adjoint code is then run with this vector as input. The output vector H\*H $\delta U$  of this adjoint code is recovered and the second scalar product is computed. The ratio of these two scalar products is then compared to the machine error (or machine epsilon), here denoted by  $\epsilon$ , which gives the upper limit of the approximation error caused by the rounding of the calculations of the machine used. The adjoint test value, here denoted by r, is therefore defined by :

$$r = \frac{\frac{\langle H\delta U, H\delta U \rangle}{\langle \delta U, H^*H\delta U \rangle} - 1}{\epsilon}$$

With the LMDz-SACS model, a valid adjoint code should not result in a r exceeding 1000 and this ratio is usually between 1 and 300 for a valid code. Adjoint tests were performed with a machine error of 2.220446049250313 ×  $10^{-16}$  (double-precision). We run two tests involving a two-month simulation based on the REF configuration. Both tests apply an increment  $\lambda$  of 0.2. The first test applied the increment in the control space (x) whereas the second does it in the minimization space ( $\chi$ =B-1/2·(x-xb)) as explained in Berchet et al. (2021).

The other configurations only modify the input data (fluxes, source signatures, prescribed errors) and do not influence the adjoint operations performed during the inversion, hence we present only the results with REF. The first test gave a ratio of 50 whereas the second test provided a ratio of 5, proving that the adjoint operations were properly implemented.

P14L16-19: Could the decreasing values of d13C in REF be also due to an underestimation of the atmospheric sink since reactions with OH and Cl enrich d13C?

**Yes, we completely agree and it has been added in the text.**

P18L20: It would be helpful if it would be stated again that this is for NOISO and REF increments.

**It has been added.**

P19L6: It is interesting that in order to correct for the prior decreasing trend in d13C, the inversion increases the source signatures of all sources, this means that the increases in the d13C rich sources, such as biofuel/biomass burning, are not sufficient to correct this trend. In T3 and T4 these emissions increased significantly, since there was not the degrees of freedom to adjust the source signatures. The question is, what is more accurate, higher source signatures or high d13C rich sources? Also, this result depends of course on having the correct atmospheric sink. Although these questions cannot be answered in this paper, I think they warrant more discussion as these are key sources of uncertainty. Also, I think the statement "All source signatures are shifted upwards by the inversions" needs to be qualified, that is, there are the exceptions of T3 and T4 (which had very small prior uncertainties for the sources signatures) and the "natural" source.

**This is a very interesting suggestion. We included some discussion about it at the end of Sect. 2.5.4. We also clarified the statement "All source signatures are shifted upwards by the inversions".**

These results must be interpreted with caution because the input data suffer from high uncertainties. The artificial increase of the source signatures by our system can be hardly related to literature and former investigations. Consequently, it is challenging to conclude whether an increase of the source signatures would be more realistic (i.e., supported by observational data) than, for instance, only increasing the emissions of 13C-enriched sources such as BB. This system is only based on a mathematical and physical framework connecting the several groups of uncertainties (observational, prior fluxes, prior source signatures, prior sinks) and finding the most likely solution. Better estimates of these uncertainties must be prescribed before obtaining robust results. In particular, the uncertainties on KIE values and sink intensities have not been tested here and could largely influence the results. Also, the prescribed uncertainties on source signatures are relatively smooth in REF compared to recent estimates (Sherwood et al., 2017). Assessing these uncertainties should be a key aspect for future studies using this new inversion system to quantify the global  $CH_4$  budget.

P19L9: I think by "total fractionation effect" the authors mean the kinetic isotope effect of atmospheric oxidation, if so, I suggest changing this to be clearer about what is meant. Also, I think it would be interesting to include a test using alternative OH fields to see how strongly the results are affected by the OH sink estimate.

This has been modified. At first, we thought about including tests using alternative OH fields (with/without inter-annual variability, with different spatial distributions, with different intensities) but we decided to limit ourselves to pure technical tests and not scientific ones in order to show only the technical potential of the system. Another study, still in preparation, is focusing on OH uncertainties with this new assimilation system. It should be submitted before the end of the year.

P19L14:18: Presumably this describes the results of the REF scenario, but it would be clearer to specify this.

**We clarified this.**

The WET global source signature, associated with REF posterior estimates, exhibits the larger upward shift compared to prior estimates...

**Reviewer #2**

**1. non-Negative Constraints**

Is there a non-negative constraint on either emissions or isotopic signatures? I doubt this since it is (or was) not easy to do in the M1QN3 algorithm used here. It is, though possible by routines in the scipy minimisation suite that still offer the same limited memory capability. The advantages can be large since a non-negative constraint removes the risk of large positive-negative flux dipoles which can inflate the posterior uncertainty.

There is no non-negative constraint on either emissions or isotopic signatures. We noticed that a highly negligible part of the posterior fluxes were negative (< 0.02 Tg/yr globally) due to the prescribed uncertainty of 100 %. This problem appears to be much less important for isotopic signatures as both positive and negative signatures can theoretically be observed, albeit positive signatures are very unlikely. The prescribed uncertainties on these signatures are, in any case, too small to lead to positive signatures in posterior estimates. We think this problem is beyond the scope of this study and discussing it in the main text could only result in an overly complicated story for the reader. It will nevertheless be addressed in future CIF developments and we are grateful to the reviewer for mentioning the scipy minimisation suite.

**Spin-up and Spin-down**

You noted on Page 18 "However, flux and source signature estimations of the 2012-2013 and 2016-2017 periods are not interpreted as the system appears to require a 2-year spin-up (2012-2013) and a 2-year spin-down (2016-2017), over which the inversion problem is not sufficiently constrained and isotopic signatures vary widely over time.". This is intriguing. It occurs, if I understand correctly, despite a long spin-up with 2012 fluxes to roughly equilibrate isotopic ratios at the start of the inversion period. Do you do this for every iteration as the control vector is up- dated? (I doubt this, it would be very expensive.) I am particularly surprised by the spin-down problem. We are used to the idea that CO 2 fluxes, at least, are only really constrained by observations a few weeks into the future. After that atmospheric mixing homogenises the Jacobian too much. Hence fluxes too close to the end of a run might lack constraint. There might be a reason why isotopic ratios would have much longer-lasting sensitivities but this isn't obvious to me and deserves some explanation.

Here, we are referring to the inversion spin-up, namely the fact that there is a potential lag between the constraint brought by the observations and the associated feedback on the fluxes / isotopic signatures. For CH4-only inversions, this spin-up and spin-down times are generally shorter than a year, if not six months. At first, we were surprised to notice such an effect but, given the very long relaxation timescales of isotopic ratios in the atmosphere (Tans et al., 1997), it seems coherent. Fully understanding this would require a lot of time and running multiple inversions (or possibly only tangent-linear simulations), starting from different initial conditions spanning the prescribed uncertainty envelope, to infer until when the initial atmospheric isotopic ratios and/or isotopic source signatures can influence the time-series of atmospheric isotopic ratios. This was too much work for this study but will certainly be addressed in future studies.

These long effects are certainly caused by the relatively long relaxation timescales of isotopic ratios in the atmosphere (Tans et al., 1997) compared to that of total  $CH_4$ . Fully understanding this would require a lot of time and running multiple inversions (or possibly only tangent-linear simulations), starting from different initial conditions spanning the prescribed uncertainty envelope, to infer until when the initial atmospheric isotopic ratios and/or isotopic source signatures can influence the time-series of atmospheric isotopic ratios. This is however beyond the scope of this study.

**Computational Cost**

The authors dwell on this a good deal. It seems almost a metric of a given set-up is its convergence rate. I suggest de-emphasising this. While I sure calculation time was frustrating it is mainly caused by the

parallellisation limits on LMDZ-SACS. If these restrictions were reduced, as they already are in some other models, this would be a less important point. It is also certain to reduce in importance as models improve.

It is a very interesting point. We agree it is mainly due to the parallelization limits of LMDz-SACS. We tried to be as comprehensive as possible on this because we think the user/reader can benefit from our experience and easily reduce the computational burden of an inversion. The S1 and T1 setups do not largely affect the results and can easily be adapted to reduce the number of iterations. We also indicated the number of hours of simulation per CPU because the inversions we performed here are relatively short compared to what would be required to rigorously investigate atmospheric methane mysteries, such as the 2000-2006 stabilization, the subsequent renewed growth or the recent large increase rates. Even with more CPUs, we suppose (hope) that the computational burden will always be a problem (decision criterion), especially if we also consider the carbon footprint of such simulation / inversion. In addition, there is a trend toward increasing the spatial resolution of CTMs in order to, for instance, assimilate a huge amount of high-resolution satellite data. It leads to a rebound effect : the more we manage to reduce computational burden, the more we want to increase spatial resolution. Finally, we also emphasized this point because we do have solutions that we mentioned in the conclusion and that could lead to an even more powerful system. For all these reasons, we think it is worth emphasizing the computational burden of the inversions performed with our system.

**Reviewer #3**

**Presentation of novelty**

As authors mention very briefly, this is a first attempt to carry out variational inversion assimilating  $\delta$ 13C(CH4) observations. Please mention this also in the abstract, and add slightly more details of the development in the Introduction, e.g. development of adjoint and implementation in CIF. From the Introduction, I was also not sure if such modelling has been done with LMDz previously, i.e. how well LMDz have been simulating  $\delta$ 13C(CH4)?

As mentioned by Reviewer #1, this is the first attempt to carry out variational inversion assimilating  $\delta^{13}C(CH_4)$  but only with a 3-D chemistry-transport model. Nevertheless, we agree with the reviewer that this has to be emphasized in the abstract and in the introduction. This is not the first official attempt to simulate  $\delta^{13}C(CH_4)$ . Another one of our papers, describing the impact of the CI sink on CH4 and  $\delta^{13}C(CH_4)$ , was previously submitted to the ACP journal (Thanwerdas et al., 2019) but was rejected, albeit not because of modeling error. We added some information in the text.

**Abstract :**

To our knowledge, this represents the first attempt to carry out variational inversion assimilating  $\delta^{13}C(CH_4)$  with a 3-D chemistry-transport model (CTM) and to independently optimize isotopic source signatures of multiple emission categories.

**Introduction :**

This new system was implemented in the Community Inversion Framework (CIF), supported by the European Union H2020 project VERIFY (http://www.community-inversion.eu) and required to implement new forward, tangent-linear and adjoint operations. The forward operations were previously used to estimate the impact of the CI sink on the modeling of  $CH_4$  and  $\delta^{13}C(CH_4)$  in LMDz (Thanwerdas et al., 2019).

**Categorization of the simulations**

I was not completely convinced about those S and T groups. Are they really needed? Did you categorize them based on results or really expected T groups to have higher variation before you started simulations? T1 is not only about changing isotope signature values and their uncertainty, but also the degree of freedom (dof) in the optimization (I guess you optimize 10 flux categories?).

We chose to use S and T groups to facilitate the presentation of results. At the beginning of the writing, no groups were used and the reading was very fastidious. We therefore decided, based on multiple feedbacks from different readers, to create the S-group using the simulations that were expected to give little variation. Based on preliminary results, it was quite easy to predict which simulations were going to be in the S-group before even running our full simulations. The remaining simulations were included in another group. We think that the reader can more easily follow the presentation of the results knowing whether a simulation has largely affected or not the results just by seeing the name of the simulation.

At the beginning, we predicted that the T1 simulation was going to be in the S-group. However, final results provided evidence that we could not reasonably include this simulation in the S-group as the variation was somehow too large for the S-group but too small for the T-group. We thought about creating a third group but it would have resulted in a presentation of results and a discussion likely difficult to read. Including the T1 simulation in the T-group but showing all the values from the T-group in Figure 6 and Figure 7 was the best compromise we found.

We recognize that this splitting may seem cumbersome at first. However, we tested many very different reading configurations and it was almost impossible to concisely and clearly present and discuss the results without these groups. We deeply apologize but we did not find any rewording, correction or addition that may resolve your concern.

**Discussion on results**

Although this technical paper is not meant to evaluate the flux estimate nor  $\delta$ 13C(CH4) values obtained from the simulations, I would like to see briefly how your estimates are compared to previous studies. Or even simply mentioning in the Conclusion how you would do further analysis, including e.g. availability of evaluation data.

We provide some additional thoughts about further analyses in the conclusion. However, we do not think this study is appropriate for comparisons with other estimates as the period 2012-2017 is never used as a period of interest in the litterature. Therefore, we prefer not to include any comparisons that could be misinterpreted. Longer periods are often selected to study the 2000-2006 stabilization and the subsequent regrowth periods. A paper, focusing on the 1998-2018 period and using the same system, is already in preparation and should be submitted by the end of the year. Some information about this new study is therefore provided in the conclusion.

As mentioned in the introduction, future work will address the estimation of  $CH_4$  emissions over longer periods of time using this new system. For instance, the 2000-2006  $CH_4$  stabilization period and the subsequent renewed growth are particularly interesting to study using the isotopic constraint as global  $\delta^{13}C(CH_4)$  started to decrease after 2006. These periods of time have already attracted considerable critical attention from many inversion studies (with or without the isotopic constraint) and comparing the results derived from such a complete 3-D variational inversion system with other recent estimates should be highly relevant. The most important limitation of assimilating  $\delta^{13}C(CH_4)$  lies in the fact that very limited  $\delta^{13}C(CH_4)$  data are available, and therefore evaluating the posterior simulated  $\delta^{13}C(CH_4)$  is often challenging, if not impossible. However, satellite and balloon / AirCore data can easily be used to evaluate the posterior simulated  $CH_4$ .

**Discussion on uncertainty estimates**

I understand that it is costly to calculate the full uncertainty from all simulations. However, you anyway present uncertainty in P12 L12. How was it calculated? From the cost function, you can speculate how dof and inclusion of additional data would affect the posterior uncertainty. Please comment on it in Section 3.5.

We suppose the reviewer is referring to page 21. We did not include much details on this part and we apologize for that. Following your recommendations, we therefore included some additional information and discussion. We calculated the full uncertainty range using the minimum and maximum values among all the configurations, as in Saunois et al. (2020). At present, this method is the only one we can use, although it is insufficient. In particular, this method does not address the fact that some configurations are less likely and less realistic than others.

The inclusion of additional  $\delta^{13}C(CH_4)$  will likely not affect the posterior uncertainty significantly because we expect the uncertainties on isotopic source signatures to have a much larger influence. We are not sure what the reviewer means by "how dof would affect the posterior uncertainty?". The posterior uncertainties of the fluxes associated to the subcategories will likely be equal or slightly larger than that of the fluxes associated to the categories but it is very difficult to speculate on this.

Formally, posterior uncertainties are given by the Hessian of the cost function. This matrix can hardly be computed at an achievable cost considering the size of the inverse problem. Other means must be implemented to get posterior uncertainty such as estimating lower-rank approximation of the Hessian, using Monte-Carlo ensembles of variational inversion to represent the prior uncertainties or computing multiple configurations covering a given range of possibilities. Here, using multiple configurations provides insight into the posterior uncertainty associated with the posterior fluxes. We calculated the full uncertainty range using the

minimum and maximum values among all the configurations, as in Saunois et al. (2020). WET, AGW, FF and BB flux estimates (Table 3) exhibit an uncertainty of 10 %, 7 %, 19 % and 38 %, respectively. BB is the most uncertain estimate relative to its intensity, although FF shows the largest absolute uncertainty (23 TgCH4.yr-1). These uncertainties are unlikely to be affected by the assimilation of additional  $\delta^{13}$ C(CH4) data because we expect the uncertainties on the isotopic source signatures to have a much larger influence. However, this remains to be tested in future work if posterior uncertainties can be calculated.

At present, M1QN3 is not the only optimization algorithm that can be utilized to perform variational inversions in the CIF. The CONGRAD algorithm (Fisher, 1998), that follows a conjugate gradient method combined with a Lanczos algorithm, is also implemented. In particular, it considerably facilitates the computation of posterior uncertainties. Any change in algorithm is very easy and accessible to any CTM embedded in the CIF. However, CONGRAD has not been tested yet with  $\delta$ 13C(CH4) data. As CONGRAD is only designed for linear problems, using this algorithm could radically change the results of inversions performed with the isotopic constraints and future work will focus on using CONGRAD to perform the inversions with isotopic constraints.

**Distribution of state vectors: did you assume all to be normal/Gaussian?**

We are not sure what the reviewer means by "distribution of state vectors". The errors associated with the control vector (can also be called state vector) and the observation vector are assumed, indeed, to be normal/Gaussian. This is already mentioned in Sect. 2.1.

**How did you derive the aggregated signature values?**

Prescribed isotopic signatures are chosen based on literature values (Text S1 in the submitted manuscript, Table 1 in the revised manuscript). Apart from wetlands, regional values (or one global value if not enough data available) are chosen and assigned on a map at LMDz spatial resolution. The 12CH4 and 13CH4 fluxes for all subcategories are then derived based on Eq. 6 and 7 and added up. The resulting fluxes are then converted back to a  $\delta^{13}C(CH_4)_{source}$  map representing the aggregated isotopic signature of the category. We included the part of this explanation that was missing in the manuscript (Sect. 2.4.1).

To infer the  $\delta^{13}C(CH_4)_{source}$  map of a category based on the sub-categories, the  ${}^{12}CH_4$  and  ${}^{13}CH_4$  fluxes for each emission sub-category within a category are derived based on Eq.5 and 6 and added up. The resulting fluxes are then converted back to a  $\delta^{13}C(CH_4)_{source}$  map representing the aggregated isotopic signature of the category.

**What is the temporal and spatial resolution of prior fluxes?**

Prior fluxes are prescribed at monthly resolution (following the EDGARv4.3.2 resolution) and at the spatial resolution of the LMDz model ( $3.8^\circ \times 1.9^\circ$ ). We added a sentence in Sect. 2.4.1 :

All prior fluxes are prescribed at monthly resolution and at the spatial resolution of LMDz.

**What is the temporal resolution of the optimization?**

**Three values per month (10 days, 10 days and the rest) for the fluxes and their associated isotopic signatures are included in the control variables. This was mentioned in Sect. 2.4.1.**

Can you provide range of observation uncertainty (diagonals of R) for each stations, maybe by adding information in Table S3 and S4, and briefly mention ranges in the main text? This will help understanding the results on cost function and RMSE differences better.

**We find this suggestion highly relevant and we added this information for each station in Table S3 and Table S4. We also added a brief sentence in the main text, in Sect. 2.4.2.**

These errors range between 3-19 ppb for CH4 observations and 0.11-0.20 ‰ for  $\delta^{13}C(CH_4)$  observations. Mean prescribed errors for each station are provided in the supplement (Tables S3 and S4).

Offsets in initial condition: How much offset did you need to add/subtract?

**We applied an offset of +1.4 ‰ to $\delta^{13}C(CH_4)$ initial conditions. It has been added to the text.**

P13 L9: "Consequently, the system is preferentially adjusting  $\delta$ 13C(CH4) over CH4 values to reduce the cost function."

• Can you speculate why? Is it because observation uncertainty (diagonals of R) is relatively smaller in  $\delta$  13 C(CH4) than CH4 ? The cost function show that the observational constraint in CH 4 is larger (probably main reason is amount of data?).

**We added some explanation / speculation to the text.**

Consequently, the system is preferentially adjusting  $\delta^{13}C(CH_4)$  over  $CH_4$  values to reduce the cost function, presumably because the ratio of RMSE to prescribed observational error for  $\delta^{13}C(CH_4)$  is, on average, about twice as large as for  $CH_4$ . In other terms, it is simpler for the system to adjust  $\delta^{13}C(CH_4)$  before attempting to modify  $CH_4$ . The ratio of the number of  $\delta^{13}C(CH_4)$  observations to the number of  $CH_4$  observations is not expected to play a significant role in the convergence process, although we did not rigorously study this influence. This ratio is only expected to affect the contribution of a component ( $\delta^{13}C(CH_4)$  or  $CH_4$ ) to the total cost function.

• For S2, I wonder why contributions of  $\delta$ 13C(CH4) and CH4 are similar to S3. Did you assimilate same amount of  $\delta$ 13C(CH4) data in REF and S2?

Yes, after the curve fitting, we sampled the pseudo-observations at the same time as the real observations. Therefore, we have the same amount of data in REF and S2. We added some explanation about the sampling following your other comment above about the curve fitting.

P14 L30-34: This could also be due to prescribed observation/transport model uncertainty.

It is likely not due to this because mean observation errors prescribed for these stations are large (10-15 ppb) but not the largest among all the assimilated stations (up to 18-19 ppb). We added this sentence to the main text.

Prescribed observation errors are likely not the main cause because mean values for these stations are large (10-15 ppb) but not the largest among all the assimilated stations. It can also be due to transport error or misrepresentation of sources close to the sites.

Emission increments: Emission changes are large in regions with high emissions. Please mention.

**We agree with this comment and added this to the main text.**

Overall, increments are large in regions with high emissions.

Please expand how much work would be needed for switching transport models and optimization methods in CIF for the  $\delta$ 13C(CH4) data assimilation. Can we use e.g. initial mixing ratios, do we need to run spin-up and

build adjoint if transport model is changed? How about changes in optimization methods? Can we use same state vectors and covariance structures?

**We included some additional information about switching transport models and optimization methods.**

The major drawback of this inversion system is undoubtedly the large computational burden of a full minimization process. At least 40 iterations appear to be necessary to reach a satisfying convergence state at the regional scale. For the LMDz-SACS model, a maximum of 8 CPUs can be run in parallel, resulting in an elapsed time of 5-6 weeks to run one of the inversions of this study. A new generation of transport models such as DYNAMICO (Dubos et al., 2015) could help to address this problem in the future by allowing more processors to run in parallel. Also, further developments will implement some parallelization methods to enable computational burden reduction (e.g., Chevallier, 2013). In addition, variational inversions as implemented in the CIF are not enabled to provide a quantification (even approximated) of the posterior uncertainties. Dedicated efforts need to be done to address this issue in the future, at an achievable numerical cost. In particular, using the CONGRAD algorithm instead of M1QN3 could be a solution as both algorithms can be easily selected in the CIF. However, additional work is needed to ensure that switching the minimization algorithm does not affect the results inferred with our new system.

This system is implemented within the CIF framework and can therefore be used for inversions with the various CTMs embedded in the CIF, provided the adjoint codes of the models exist. As the operations developed for the purpose of this study are performed outside the model structure, forward, tangent-linear and adjoint codes and adjoint codes from other CTMs do not require any modifications as long as the model is capable of simulating both 12CH4 and 13CH4 simultaneously. The prior input must be adapted to the new model (spatial and time resolution) but the format of the observational data and of the prescribed errors can be preserved. Also, due to the variational method benefits, the efforts dedicated to the preparation of inputs do not scale with either the size of the observational datasets or the length of the simulation time-window. Therefore, this system is very powerful and is particularly relevant to study in a consistent way the influence of multiple physical parameters on atmospheric isotopic ratios, such as the transport, the isotopic signatures, the emission scenarios, the KIE values, etc. We did not try to assess here the sensitivity of the system to these parameters as only technical aspects of the system were tested. This will be part of future analyses.

**Figure 3: Please add label of x-axis**

**This has been done.**

**Figure 4: Please add legend of posterior results from REF simulation, and perhaps use different color than green, as it's not S-group simulation? Please also add results from NOISO.**

We added the legend associated with the REF simulation and applied a different color. For the  $\delta^{13}C(CH_4)$  panel, posterior NOISO global source signature is -54.1 ‰ and the NOISO line would reach lower values than PRIOR REF in 2017, resulting in an image even more zoomed out and therefore affecting the clarity. We therefore do not suggest to include NOISO but we mention it in the caption. For CH4, the line is actually extremely close to the REF line and it would also affect the clarity to include it. We also include this explanation in the caption.

The NOISO lines were not included because 1) the posterior NOISO global source signature is -54.1 ‰ and the line would therefore reach lower values than the REF PRIOR, affecting the visual clarity of the upper plot. 2) The NOISO  $CH_4$  values are extremely close to the REF values and including it would also affect the clarity of the lower plot.

Figure 4 caption: I guess the figure is global monthly mean?

**Yes. We modified the caption.**

**Figure 5: Prior CH 4 is same for all simulations, and $\delta$ 13 C(CH 4) for some. Please consider minimizing.**

We strongly recommend not to minimize the plot. A lower number of boxes in only one or two panels would force us to adapt the x-labels and the box colors depending on the panel. We think that it would strongly affect the visual clarity of the plot. We nevertheless reduced the number of station labels in the lower-left panel as they are all the same.

*Figure S2: Please add label and unit of y-axis. Caption is slightly unclear – what do you mean by "inferred with REF"?*

We added the label and modified the caption to make it clearer.

**Technical comments**

P3L2: constrain -> constraint This has been modified.

P3L3: have -> has This has been modified.

P3L6: regrowth -> renewed growth This has been modified.

P4L16: This phrase is not grammatically correct, please change to "allowing for the inverse to be calculated easily" This has been modified.

P5L18: multi-constrain -> multi-constraint This has been modified.

P12L3-L16: This would be easier to follow if the list items (i.e. the different inversion tests) would be numbered. **This has been modified.**

P19L6: source signature -> source signatures **This has been modified.**

P21L16: relatively -> relative This has been modified.

P14 L22: The S-group provides a better match to  $\delta$ 13C(CH4) observations than... **This has been modified.**

P15 L4-5: AMY is not in South-East Asia. We changed this to "South-East and East Asia".

**Variational inverse modelling modeling within the Community Inversion Framework v1.1 to assimilate $\delta^{13}C(CH_4)$ and $CH_4$ : a case study with model LMDz-SACS**

Joël Thanwerdas1,\*, Marielle Saunois1, Antoine Berchet1, Isabelle Pison1, Bruce H. Vaughn2, Sylvia Englund Michel2, and Philippe Bousquet1

1Laboratoire des Sciences du Climat et de l'Environnement, CEA-CNRS-UVSQ, IPSL, Gif-sur-Yvette, France. 2INSTAAR - University of Colorado, Boulder, CO, United States

Correspondence: J. Thanwerdas (joel.thanwerdas@lsce.ipsl.fr)

**Abstract.**

Atmospheric  $CH_4$  mixing ratios  $CH_4$  mole fractions resumed their increase in 2007 after a plateau during the period 1999-2006, suggesting varying period, indicating relative changes in the sources and sinksas main drivers. Estimating sources by exploiting observations within an inverse modeling framework (top-down approaches) is a powerful approach. It is neverthe-

- 5 less challenging to efficiently differentiate co-located emission categories and sinks by using  $CH_4$  CH4 observations alone. As a result, top-down approaches are limited when it comes to fully understanding  $CH_4$  CH4 burden changes and attribute these changes to specific source variations.  $CH_4$ -source isotopic signatures  $\delta^{13}C(CH4)_{source}$  isotopic signatures of CH4 sources differ between emission categories (biogenic, thermogenic and pyrogenic), and can therefore be used to address this limitation. Here, a new 3-D variational inverse modeling framework designed to assimilate  $\delta^{13}C(CH_4)$ - $\delta^{13}C(CH_4)$  observations
- 10 together with  $CH_4$ -CH4 observations is presented. This system is capable of optimizing both the emissions and associated source signatures of multiple emission categories at the pixel scale. To our knowledge, this represents the first attempt to carry out variational inversion assimilating  $\delta^{13}C(CH_4)$  with a 3-D chemistry-transport model (CTM) and to independently optimize isotopic source signatures of multiple emission categories. We present the technical implementation of joint  $CH_4$ and  $\delta^{13}C(CH_4)$  CH4 and  $\delta^{13}C(CH_4)$  constraints in a variational system, and analyze how sensitive the system is to the setup
- 15 controlling the optimization using the 3-D Chemistry-Transport Model LMDz-SACS 3-D CTM. We find that assimilating  $\delta^{13}C(CH_4)$   $\delta^{13}C(CH_4)$  observations and allowing the system to adjust source isotopic isotopic source signatures provide relatively large differences in global flux estimates for wetlands (5-), microbial (6-5.7 TgCH4.yr-1), agriculture and waste (-6.4 TgCH4.yr-1), fossil fuels (8+8.6 TgCH4.yr-1) and biofuels-biomass burning (4+3.2 TgCH4.yr-1) categories compared to the results inferred without assimilating  $\delta^{13}C(CH_4)$   $\delta^{13}C(CH_4)$  observations. More importantly, when assimilating
- 20 both  $CH_4$  and  $\delta^{13}C(CH_4)$ -CH4 and  $\delta^{13}C(CH_4)$  observations, but assuming that the source signatures are perfectly known, increase these differences between the system with  $CH_4$  and the enhanced one with  $\delta^{13}C(CH_4)$  these differences increase by a factor 3 or 4 of 3-4, strengthening the importance of having as accurate as possible signatures signature estimates as possible. Initial conditions, uncertainties on  $\delta^{13}C(CH_4)$  in  $\delta^{13}C(CH_4)$  observations or the number of optimized categories have a much smaller impact (less than 2 TgCH4.yr-1).

**25 1 Introduction**

Methane ( $CH_4CH_4$ ) is a powerful greenhouse gas and is responsible for 23 % (Etminan et al., 2016) of the radiative forcing induced by the well-mixed greenhouse gases ( $CO_2$ ,  $CH_4CH_4$ ,  $N_2O$ ). Atmospheric  $CH_4$  mixing ratios  $CH_4$  mole fractions have increased quasi-continuously since the pre-industrial era and by about 9 ppb/yr ppb.yr-1 from 1984 to 1998 (www.esrl.noaa.gov/gmd/ccgg/trends\_ch4/)(Dlugokencky, 2021). After a plateau between 1999 and 2006 that still generates

- attention and controversy (e.g., Fujita et al., 2020; Thompson et al., 2018; McNorton et al., 2018; Turner et al., 2017; Schaefer et al., 2016; Schwietzke et al., 2016; Rice et al., 2016), the mixing ratios mole fractions resumed their increase at a large rate, exceeding 10 ppb/yr-ppb.yr-1 in 2014 and 2015. Trends in atmospheric  $CH_4$  CH4 are caused by a small imbalance between large sources and sinks. Assessing their spatio-temporal characteristics is particularly challenging considering the variety of methane CH4 emissions. Yet, identifying and quantifying the processes contributing to these changes is mandatory to formu-
- 35 late relevant  $CH_4$  CH4 mitigation policies that would contribute to meet the target of the 2015 UN Paris Agreement on Climate Change and to limit climate warming to 2°C °C.

Thanks to continuous efforts of surface monitoring networks, the spatial coverage and the accuracy of the atmospheric methane- $CH_4$  measurements provided to the scientific community increased over the last decades. Consequently, top-down estimates using inversion methods emerged and became relevant, along with bottom-up estimates, to explain and quantify the

- 40 recent sources and sinks variations. The first inverse modeling techniques were designed in the late 1980s and early 1990s for inferring greenhouse gas sources and sinks from atmospheric CO2 measurements (Enting and Newsam, 1990; Newsam and Enting, 1988). The Without regularization of the problem, e.g. providing prior information, the inverse problem is considered as "ill-conditioned (or ill-posed" (non-uniqueness of the solution, no continuity with the data) and therefore necessitates as many constraints as possible to be regularized. Several ). It means that there is no unique solution to the problem but also
- 45 that a small error in the assimilated data (here atmospheric observations) can result in large errors in the derived solution. Several inversion methods have been designed over the years, among which analytical (e.g., Bousquet et al., 2006; Gurney et al., 2002), ensemble (e.g., Zupanski et al., 2007; Peters et al., 2005) and variational methods (e.g., Chevallier et al., 2005). The variational formulation uses the adjoint equations of a specific model to compute the gradient of a cost function and then minimize it, for example using a gradient descent method. Computational times and memory costs do not scale with the
- 50 number of measurements and the number of variables to control, contrary to the analytical and ensemble methods, which can hardly accommodate very large observational datasets and control vectors at the same time. Thus, the variational formulation is preferred to the others when optimizing emissions and sinks at the pixel scale using large volumes of observational data, although its main limitation is the numerical cost to access posterior uncertainties when there is non-linearity in the inversion problem (Berchet et al., 2021).
- 55 Inversion systems generally assimilate measurements from ground-based stations and/or satellites to constrain the global sources and sinks of CH4CH4, starting from a prior knowledge of these. These systems are very effective to provide total emission estimates (e.g., Saunois et al., 2020; Bergamaschi et al., 2018, 2013; Saunois et al., 2017; Houweling et al., 2017, and references therein). However, differentiating the contributions of multiple co-located CH4 CH4 source categories is challenging

as it only relies on different seasonality cycles and on applied spatial distributions and error correlations (e.g., Bergamaschi

60 et al., 2013, 2010). The atmospheric isotopic signal contains additional information on methane-CH4 emissions that can help to separate emission categories based on their source origin. The atmospheric isotopic signal  $\delta^{13}C(CH_4)\delta^{13}C(CH_4)$  is defined as:

$$\underline{\delta^{13}C(CH_4)}\delta^{13}C(CH_4) = \frac{R}{\frac{R_{std}}{R_{std}}}\frac{R}{\frac{R_{std}}{R_{std}}} - 1$$
(1)

where R and  $R_{std}$  std denote the sample and standard  $\frac{^{13}CH_4:^{12}CH_4}{^{13}CH_4:^{12}CH_4}$  ratios. We use the VPDB-Vienna -

- 65 Pee Dee Belemnite (V-PDB) scale with  $R_{\overline{std}\_std\_}=0.00112372$  (Craig, 1957) throughout this paper.  $CH_4$  source isotopic signatures  $\delta^{13}C(CH_4)_{source\_}$  The isotopic source signatures of  $CH_4$ , here denoted by  $\delta^{13}C(CH_4)_{source\_}$ , notably differ between emission categories ranging from 13C-depleted biogenic sources (approx. -62 ‰ -61.7 ± 6.2 ‰, one standard deviation) and thermogenic sources (approx. -44 ‰ -44.8 ± 10.7 ‰) to 13C-enriched thermogenic sources (approx. -22 ‰) (Sherwood et al., 2017; Schwietzke et al., 2016) - 26.2 ± 4.8 ‰) (Sherwood et al., 2017; Schwietzke et al., 2016), although
- 70 the distributions are very large and overlaps exist between the extreme values. Consequently,  $\delta^{13}C(CH_4) \delta^{13}C(CH_4)$  depends on both the CH4 CH4 emissions and their isotopic signatures. Saunois et al. (2017) pointed out that many emission scenarios inferred from atmospheric inversions are not consistent with  $\delta^{13}C(CH_4) \delta^{13}C(CH_4)$  observations and that this constrain constraint must be integrated into the inversion systems to avoid such inconsistencies. In addition, they highlighted the sensitivity of the atmospheric isotopic signal to the source partitioning and prescribed isotopic ratios. Since the 1990s,
- 75  $\delta^{13}C(CH_4)$  have  $\delta^{13}C(CH_4)$  has been monitored at multiple sites, although less than for total CH4, providing opportunities to use this constraint within an inversion framework. In addition, these values have been shifting towards smaller more negative values since 2006 (Nisbet et al., 2019) when CH4 CH4 trends resumed their increase, suggesting that this isotopic data can help to understand the processes that contributed to the regrowthrenewed growth. However, implementing the assimilation of such measurements into an inversion system is not straightforward and introduces additional complexity.
- Hereinfafter, the assimilation of  $\delta^{13}C(CH_4)$ - $\delta^{13}C(CH_4)$  observations to constrain the estimates of an inversion is referred to as the "isotopic constraint". The implementation of such a constraint in an inversion system have already been attempted in previous studies focusing on CH4-CH4 (e.g., Thompson et al., 2018; McNorton et al., 2018; Rigby et al., 2017; Rice et al., 2016; Schaefer et al., 2016; Schwietzke et al., 2016; Rigby et al., 2012; Neef et al., 2010; Bousquet et al., 2006; Fletcher et al., 2004) but, to our knowledge, never in a variational system associated to a 3-D chemistry-transport model (CTM). Adding this
- 85 isotopic constraint to a variational inversion system is challenging as, in contrast to an analytic inversion in which the response functions of the model are precomputed, the isotopic constraints have to be considered both in the forward (simulated isotopic values) and the adjoint (sensitivity of isotopic observations to optimized variables) versions of the model.

[revised manuscript text omitted]

- The offline model LMDz is coupled with the Simplified Atmospheric Chemistry System (SACS) (Pison et al., 2009). This chemistry system was previously used to simulate the oxidation chain of hydrocarbons, including  $CH_4CH_4$ , formaldehyde (CH2O), carbon monoxide (CO) and molecular hydrogen (H2)together with methyl chloroform (MCF). For the purpose of this
- study, this system has been was converted into a chemistry parsing system. It follows the same principle as the one used by the regional model CHIMERE (Menut et al., 2013) and therefore allows for user-specific chemistry reactions. As a result, it generalizes the previous SACS module to any possible set of reactions. The adjoint code has also been implemented to allow variational inverse modellingmodeling. The different species are either prescribed (here OH, O(1D) and Cl) or simulated (here  $\frac{12}{CH_4}$  and  $\frac{13}{CH_4}$  CH4 and  $\frac{13}{CH_4}$ ). The prescribed species are not transported in LMDz, nor are their mixing ratios mole
- 150 fractions updated through chemical production or destruction. Such species are only used to calculate reaction rates to update simulated species at each model time step. In this study, the isotopologues  ${}^{12}CH_4$  and  ${}^{13}CH_4$  and  ${}^{13}CH_4$  are simulated as separate tracers and  $CH_4$  CH4 is defined as the sum of both isotopologues. Cl +  $CH_4$  oxidation has been CH4 oxidation was implemented to complete the chemical removal of  $CH_4$ CH4, which previously only accounted for OH +  $CH_4$  CH4 and  $O({}^{1}D)$ +  $CH_4$ -CH4 in the SACS scheme. Fractionation values (KIE for-
- 155

In the atmosphere, radicals (OH, O(1D) or Cl) react faster with  $^{12}$ CH4 than with  $^{13}$ CH4. This effect is called the Kinetic Isotope Effect +(KIE) or the fractionation effect. Fractionation values are prescribed to the different sinks in SACS. Here, KIE

this value is defined by  $\text{KIE} = k_{12}/k_{13}$  where  $k_{12}$  is the constant rate of the reaction involving 12CH4 and rate constant of a reaction between a radical and 12CH4,  $k_{13}$  is the constant rate of the same reaction involving 13CH4 rate constant of the reaction between the same radical and 13CH4. Additional information is and prescribed KIE values are provided in the supplement (Text S2).

The chemistry-transport LMDz-SACS is used to test the new variational inverse modelling modeling system that is described in the next section.

**2.3 Technical implementation of the isotopic constraint**

The isotopic multi-constrain multi-constraint system was implemented in the Community Inversion Framework (CIF),
supported by the European Union H2020 project VERIFY (http://www.community-inversion.cu)CIF. The CIF has been designed to allow comparison of different approaches, models and inversion systems used in the inversion community (Berchet et al., 2020)(Berchet et al., 2021). Different atmospheric transport models, regional and global, Eulerian and Lagrangian are implemented within the CIF. The system presented in this paper has been originally designed to run and be tested with LMDz-SACS but can theoretically be coupled with all models implemented in the CIF framework. The system is
able to :

- Assimilate  $\frac{\delta^{13}C(CH_4)}{\delta^{13}C(CH_4)}$  and  $CH_4$   $\delta^{13}C(CH_4)$  and  $CH_4$  observations together.
- Independently optimize fluxes and isotopic signatures for multiple emission categories.
- Optimize  $\frac{\delta^{13}C(CH_4)}{\delta^{13}C(CH_4)}$  and  $CH_4$   $\delta^{13}C(CH_4)$  and  $CH_4$  initial conditions.

Figure 1–1 shows the different steps of a minimization iteration of the cost function. Each iteration performed with the 175 descent algorithm can be decomposed into four main steps presented below. For clarity, we only present here the optimization of  $CH_4$  CH4 fluxes and associated source signatures but  $CH_4$  and  $\delta^{13}C(CH_4)$  CH4 and  $\delta^{13}C(CH_4)$  initial conditions can also be optimized by the system following the same process.

1. The process starts with a forward run. The different flux variables are extracted and converted into  $\frac{^{12}CH_4}{^{13}CH_4}$  and  $^{13}CH_4$  mass fluxes for each category following the Eq.(5)-(7) below. 5 and 6 below.

180
$$\underline{A^{i} = (1 + \delta^{13}C(CH_{4})^{i}_{\text{source}}) \cdot R_{std}} F_{12}^{i} = \frac{M_{12}}{\underline{M_{TOT}}} \frac{M_{12}}{M_{T}} \cdot \frac{1}{1 + A^{i}} \cdot F_{\underline{TOTT}}^{i}$$
(5)

$$F_{13}^{i} = \frac{M_{13}}{\underline{M_{TOT}}} \frac{M_{13}}{\underline{M_{T}}} \cdot \frac{A^{i}}{1+A^{i}} \cdot F_{\underline{TOTT}}^{i}$$
(6)

with

160

$$\underline{A^{i} = (1 + \delta^{13} \mathrm{C}(\mathrm{CH4})^{i}_{\mathrm{source}}) \cdot R_{\mathrm{std}}}$$
(7)

 $F_{TOT}^{i} F_{T}^{i}$ ,  $F_{12}^{i}$  and  $F_{13}^{i}$  are the CH4, 12CH4 and 13CH4 fluxes in CH4, 12CH4 and 13CH4 mass fluxes of a specific category *i*, respectively.  $M_{TOT}M_{T}$ ,  $M_{12}$  and  $M_{13}$  are the CH4, 12CH4 and 13CH4 CH4, 12CH4 and 13CH4 molar masses, respectively.  $\delta^{13}$ C(CH4) $\delta^{13}$ C(CH4)*i*source is the source-isotopic signature of the category *i*.  $M_{T}$  should preferably depend on  $M_{12}$  and  $M_{13}$  when converting the mass fluxes:

$$M_T = \frac{M_{12} + A^i \cdot M_{13}}{1 + A^i}$$
(8)

However, the complexity of the forward, tangent-linear and adjoint codes would be largely enhanced by such a relationship. The code structure would also be less generic, i.e., it could not be used for a joint assimilation of multiple isotopologues of CH4, such as both  $\delta^{13}$ C(CH4) and  $\delta$ D(CH4). We choose to implement  $M_T$  as a constant that can be prescribed freely by the user, therefore without considering any influence of the  $M_{12}$  and  $M_{13}$  values, also prescribed by the user. As the observed isotopic source signatures roughly vary between -70% and -10%, a maximum variation of 0.004 % in  $M_T$  could be expected. It will very likely not affect the results of our study or that of any other inversion performed with our system.

The  ${}^{12}\text{CH}_4$  and  ${}^{13}\text{CH}_4$   ${}^{12}\text{CH}_4$  and  ${}^{13}\text{CH}_4$  total fluxes are then provided calculated by summing all categories and used by the model LMDz-SACS to simulate the  ${}^{12}\text{CH}_4$  and  ${}^{13}\text{CH}_4$  atmospheric mixing ratios  ${}^{12}\text{CH}_4$  and  ${}^{13}\text{CH}_4$  atmospheric mole fractions over the time-window considered. FinallyAfter the simulation, the simulated values are converted back to  ${}^{\text{CH}_4}$  and  ${}^{\delta^{13}}\text{C}(\text{CH}_4)$  to  ${}^{\text{CH}_4}$  and  ${}^{\delta^{13}}\text{C}(\text{CH}_4)$  simulated equivalent of the assimilated observations using Eq.(8) and(9) below : 9 and 10 below :

$$[CH_4CH_4] = [{}^{12}CH_4{}^{12}CH_4] + [{}^{13}CH_4{}^{13}CH_4]$$
(9)

$$\underline{\delta^{13}C(CH_4)}\delta^{13}C(CH_4) = \frac{\begin{bmatrix} 1^{13}CH_4 \end{bmatrix}}{\begin{bmatrix} 1^{12}CH_4 \end{bmatrix}} \frac{\begin{bmatrix} 1^{13}CH_4 \end{bmatrix}}{\begin{bmatrix} 1^{12}CH_4 \end{bmatrix}} \cdot \frac{1}{\underline{R_{std}}} \frac{1}{\underline{R_{std}}} - 1$$
(10)

 $[CH_4], [^{12}CH_4]$  and  $[^{13}CH_4]$  are CH4,  $^{12}CH_4$  and  $^{13}CH_4$  atmospheric mixing ratios  $[CH_4], [^{12}CH_4]$  and  $[^{13}CH_4]$  are CH4,  $^{12}CH_4$  and  $^{13}CH_4$  atmospheric mole fractions simulated by the model in mol mol-1, respectively.

205 2. These simulated values are then compared to the available observations in order to compute  $\mathcal{H}(\mathbf{x}) - \mathbf{y}^{\circ} - \mathcal{H}(\mathbf{x}) - \mathbf{y}^{\circ}$ which is further used to infer the cost function and generate  $CH_4$  and  $\delta^{13}C(CH_4)$   $CH_4$  and  $\delta^{13}C(CH_4)$  adjoint forcings (indicated by the "\*" star superscript symbol) that compose the vector  $\delta \mathbf{y}^*$ :

$$\delta \mathbf{y}^* = \mathbf{R}^{-1}(\mathcal{H}(\mathbf{x}) - \mathbf{y}^{\mathbf{o}}) \tag{11}$$

This Although this vector is normally used directly as input to the adjoint model (see Eq.4)but in the new system, the adjoint forcings CH4 - 4), the CH4 and  $\frac{^{13}C(CH_4)}{^{13}C(CH_4)}\delta^{13}C(CH_4)$  adjoint forcings must first be converted into the adjoint forcings  $^{12}CH_4$   $^{12}CH_4$  and  $\frac{^{13}CH_4}{^{13}CH_4}\delta^{13}C(CH_4)$  adjoint forcings in the new system.

7

200

210

185

190

195

3. The newly designed adjoint code that converts  $CH_4$  and  $\delta^{13}C(CH_4)$   $CH_4$  and  $\delta^{13}C(CH_4)$  adjoint forcings into  $\frac{^{12}CH_4}{^{13}CH_4}$   $\frac{^{13}CH_4}{^{12}CH_4}$   $\frac{^{13}CH_4}{^{12}CH_4}$   $\frac{^{13}CH_4}{^{12}CH_4}$  adjoint forcings is based on the Eq.(11)-(13)-12, 13 and 14 depending on the type of the initial observation.

215
$$[\underline{{}^{12}CH_4}{}^{12}CH_4]^* \underline{{}^{CH_4}CH_4} = [\underline{{}^{13}CH_4}{}^{13}CH_4]^* \underline{{}^{CH_4}CH_4} = [\underline{CH_4}CH_4]^*$$
(12)

$$[\underbrace{{}^{12}CH_4}{}^{12}CH_4]^* \underbrace{{}^{13}C\delta^{13}C}_{\underline{\delta}^{13}C} = -\underbrace{\frac{[{}^{13}CH_4]}{[{}^{12}CH_4]^2}}_{\underline{[{}^{12}CH_4]^2}} \cdot \underbrace{\frac{1}{R_{std}}}_{\underline{R_{std}}} \cdot \underbrace{\frac{1}{\delta}^{13}C(CH_4)}_{\underline{\delta}^{13}C(CH_4)} \delta^{13}C(CH_4)^*$$
(13)

$$[\underline{{}^{13}CH_4}^{13}CH_4]^* \underline{\delta}^{13}C \delta \underline{\delta}^{13}C = \underline{\frac{1}{[{}^{12}CH_4]}} \underbrace{\frac{1}{[{}^{12}CH_4]}} \cdot \underbrace{\frac{1}{R_{std}}}{\frac{1}{R_{std}}} \cdot \underbrace{\frac{\delta}{\delta}^{13}C(CH_4)}^{13} \delta^{13}C(CH_4)^*$$
(14)

 $\frac{[^{12}CH_4]^*_{CH_4}}{[^{12}CH_4]^*_{CH_4}} \frac{[^{12}CH_4]^*_{CH_4}}{[^{12}CH_4]^*_{CH_4}} \text{ and } \frac{[^{13}CH_4]^*_{CH_4}}{[^{12}CH_4]^*_{\delta^{13}C}} \text{ are adjoint forcings associated with } \frac{[^{12}CH_4]^*_{\delta^{13}C}}{[^{12}CH_4]^*_{\delta^{13}C}} \frac{[^{12}CH_4]^*_{\delta^{13}C}}{[^{12}CH_4]^*_{\delta^{13}C}} \frac{[^{12}CH_4]^*_{\delta^{13}C}}{[^{12}CH_4]^*_{\delta^{13}C}} \text{ and } \frac{[^{13}CH_4]^*_{\delta^{13}C}}{[^{12}CH_4]^*_{\delta^{13}C}} \text{ are adjoint forcings associated with } \frac{\delta^{13}C(CH_4)}{\delta^{13}C(CH_4)} \delta^{13}C(CH_4) \text{ observations. The adjoint code of the CTM is then run with these adjoint forcings as inputs.}$

Outputs of the adjoint run provide the sensitivities of the adjoint forcings to the  ${}^{12}$ CH4 and  ${}^{13}$ CH4 and  ${}^$

$$F_{\underline{TOTT}}^{*,i} = \frac{1}{1+A} \cdot \left[\frac{M_{12}}{M_{TOT}} \frac{M_{12}}{M_{T}} \cdot F_{12}^{*,i} + \frac{M_{13}}{M_{TOT}} \frac{M_{13}}{M_{T}} \cdot A \cdot F_{13}^{*,i}\right]$$
(15)

$$\underline{\delta^{13}C(CH_4)_{\text{source}}^{*,i}} \delta^{13}C(CH_4)_{\text{source}}^{*,i} = R_{\underline{stdstd}} \cdot \underbrace{\frac{F_{TOT}}{(1+A)^2} \frac{F_T}{(1+A)^2}}_{\underline{(1+A)^2}} \cdot [\underbrace{\frac{M_{13}}{M_{TOT}} \frac{M_{13}}{M_T}}_{\underline{M_{TOT}}} \cdot F_{13}^{*,i} - \underbrace{\frac{M_{12}}{M_{TOT}} \frac{M_{12}}{M_T}}_{\underline{M_{TOT}}} \cdot F_{12}^{*,i}]$$
(16)

4. The minimization algorithm uses utilizes these sensitivities to compute the gradient of the cost function. It then finds an optimized control vector that reduces reducing the cost function and that is used for the next iteration.

In order to confirm that the several adjoint operations have been correctly implemented, we also provide the results of multiple adjoint tests in the supplement (Text S4).

**2.4 Setup of the reference simulation**

220

225

The reference configuration (REF) is a variational inversion that optimizes the  $CH_4$ -CH4 emission fluxes and  $\delta^{13}C(CH_4)$ source isotopic  $\delta^{13}C(CH_4)$  isotopic source signatures of five different categories (: biofuels-biomass burning , microbial(BB), agriculture and waste (AGW), fossil fuels , natural and wetlands ) (FF), wetlands (WET) and other natural sources (NAT). CH4

and the  $CH_4/\delta^{13}C(CH_4)$  initial conditions  $\delta^{13}C(CH_4)$  initial conditions are also optimized. The assimilation time-window is the period 2012-2017 period. The five categories originate from an aggregation of ten sub-categories (Table 1) and see Table 1) and are chosen to be as isotopically consistent as possible. Sinks are not optimized here.